# No evidence that Chinese playtime mandates reduced heavy gaming in one segment of the video games industry

David Zendle [1] ✉, Catherine Flick [2], Elena Gordon-Petrovskaya [1], Nick Ballou [3], Leon Y. Xiao [4] & Anders Drachen [1,5]

Governments around the world are considering regulatory measures to reduce young people's time spent on digital devices, particularly video games. This raises the question of whether proposed regulatory measures would be effective. Since the early 2000s, the Chinese government has been enacting regulations to directly restrict young people's playtime. In November 2019, it limited players aged under 18 to 1.5 hours of daily playtime and 3 hours on public holidays. Using telemetry data on over seven billion hours of playtime provided by a stakeholder from the video games industry, we found no credible evidence for overall reduction in the prevalence of heavy playtime following the implementation of regulations: individual accounts became 1.14 times more likely to play heavily in any given week (95% confidence interval 1.139–1.141). This falls below our preregistered smallest effect size of interest (2.0) and thus is not interpreted as a practically meaningful increase. Results remain robust across a variety of sensitivity analyses, including an analysis of more recent (2021) adjustments to playtime regulation. This casts doubt on the effectiveness of such state-controlled playtime mandates.

Playing video games is a common human activity: industry estimates suggest that more than two billion individuals now spend hundreds of billions of hours playing video games every year[1,2]. In parallel to the mass adoption of video gaming as a leisure pursuit, we have seen the rise of concerns regarding excessive engagement with video games[3,4], which have become part of the wider public debate about the health impacts of 'screen time'[5]. In the wake of these concerns, various governments have considered regulatory measures aimed at reducing playtime, particularly among young people. The most radical of these was enacted in Mainland China in 2019: the 'Notice on the Prevention of Online Gaming Addiction in Juveniles' mandated that individuals aged under 18 played no more than 1.5 hours each day (or 3 hours on public holidays)[6]. Despite the importance of this regulation, its effectiveness

has previously been impossible to establish due to a lack of large-scale behavioural data regarding playtime in China. Here we use approximately seven billion hours of playtime data drawn from Mainland China in the weeks preceding and following the implementation of playtime mandates to investigate whether these regulations were effective in reducing heavy gaming.

For video games, debates regarding excessive use typically centre around the idea that some individuals may over-engage with video games in a dysregulated manner that is similar in symptomatology and outcome to substance use disorders[7], with further negative impacts on players' physical, mental, social and financial wellbeing. In this vein, in 2013, the American Psychiatric Association identified 'Internet Gaming Disorder' as a prospective condition for future study; in 2019, the

[1]Department of Computer Science, University of York, York, UK. [2]School of Computer Science and Informatics, De Montfort University, Leicester, UK. [3]School of Electronic Engineering and Computer Science, Queen Mary University of London, London, UK. [4]Center for Digital Play, IT University of Copenhagen, Copenhagen, Denmark. [5]SDU Metaverse Lab, Maersk McKinney-Moeller Institute, University of Southern Denmark, Odense, Denmark. ✉e-mail: david.zendle@york.ac.uk

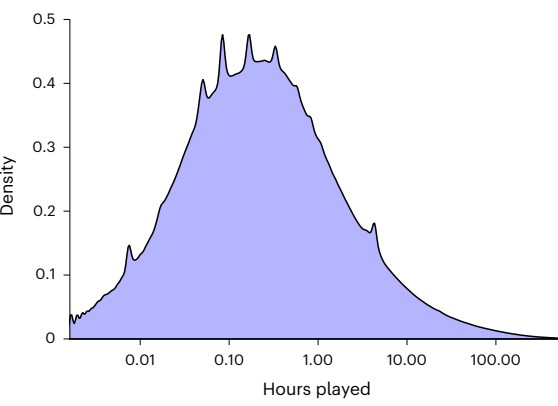

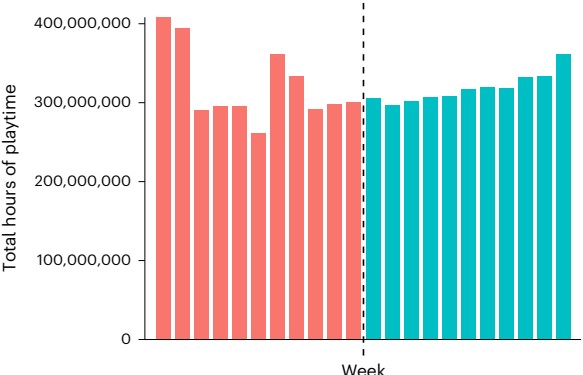

**Fig. 1 | Summary of dataset 1.** The graph on the left shows the density of hours played per gamer: this visualization is based on data from a random subsample of 100 million accounts drawn from dataset 1. The majority of individuals in our dataset played for a total of less than 1 hour during the period in question, as would be expected from a cross-sectional dataset of mobile gameplay[83].

Due to the heavily log-normal nature of the data distribution, the *x* axis is log-transformed. The chart on the right shows the total number of hours of playtime in our dataset, split per week. The dashed line represents the implementation of regulations on 1 November 2019.

World Health Organization designated 'gaming disorder' as a clinical condition in the *International Classification of Diseases: 11th Revision* (ICD-11)[8,9].The validity of conditions such as gaming disorder (ICD-11) and Internet Gaming Disorder (*Diagnostic and Statistical Manual of Mental Disorders: Fifth Edition*)[9] is heavily contested. The potential codification of Internet Gaming Disorder in the *Diagnostic and Statistical Manual* attracted substantial controversy and criticism[10,11]. Similarly, the World Health Organization's decision to add gaming disorder into the ICD-11 has led to widespread debate among academics regarding its appropriateness[12,13]. Indeed, research on dysregulated gaming in general is characterized by a lack of consensus. Some scholars argue that it constitutes a highly prevalent behavioural addiction linked to social isolation, psychopathology and low life satisfaction, and, as such, constitutes a major public health issue[14–16]. Other scholars contest that the screen time and dysregulated gaming discourses are but the most recent 'technology panic', founded on biased, low-quality evidence[17]. They argue that clinically important distress around gaming is far less prevalent than current self-report scales suggest; and that heavy gaming is likely not a genuine disorder, but rather a coping strategy or symptom of some other underlying social or mental issues[14,18–21]. These latter scholars point to recent large-scale studies that suggest analytic flexibility can produce anything from a positive to a null to a negative correlation between playtime and wellbeing; and that a person's total playtime does not predict substantial variance in wellbeing[18,22]. They further argue that video game play may have positive wellbeing impacts, such as recovering from negative experiences, improving emotional regulation or relieving stress[23–25].

Part and parcel of this contested debate is the lack of consensus on what constitutes 'heavy' or 'dysregulated' gaming. The current literature offers little guidance about how and why to segment the gaming population into 'heavy' and 'non-heavy' subgroups[26–28]. In one study, researchers define heavy gaming as 2 or more hours of playtime per day[29]. In another, heavy players are those who play for more than 30 hours per week[30]. One plausible proposed segmentation scheme is given by Colder-Carras et al.[31] Noting the lack of consensus in the literature, they suggest that heavy play may be defined as an individual spending more than 4 hours per day, 6 days per week in-game—a cut-off which they base on alignment with both clinical qualitative findings and local and international population samples, citing, for example, how focus groups with clinicians have suggested that spending more than 4 hours per day playing games may be a sign of disordered play[32].

Furthermore, there is evidence to suggest that how people engage in 'heavy' gaming, however defined, moderates its impact on wellbeing[33]. Thus, Karhulahti et al. found through interpretative phenomenological analysis of interviews that 'experiences of disorder derive from gaming interfering with what one wants to be, do, and have throughout life, whereas the experiences of intensive esport play derive from gaming being integrated into self throughout life'[34]. Thus, even if a cut-off for heavy gaming were universally accepted, heavy gaming in and of itself may not be able to reliably predict health[18,22].

Regardless of this ongoing academic debate over dysregulated gaming, several governments around the world have put it on their policy and regulatory agenda[35,36], particularly in East Asia[37,38]. In 2003, Thailand imposed a night curfew on online gaming. In 2011, the Vietnamese government banned online gaming between 22:00 and 08:00. In South Korea, the recently repealed 'Cinderella law' prohibited online game companies from providing services to individuals aged under 16 between midnight and early morning for the decade spanning 2011 to 2021[39–42].

The most consistent and restrictive governmental regulation of play, however, has been occurring in Mainland China. From 2000 onwards, the Chinese government has variously restricted the production, import and sale of gaming consoles and arcade machines (such practices were initially restricted, restrictions were later repealed and practices were later officially permitted[43–45]); mandated online game providers to install 'anti-addiction software'[6,46]; and repeatedly paused the government approval process for new video gaming licences, which every game title needs to be legally available[47]. Effective November 2019, the Chinese government enacted a further policy controlling access to gaming among young people. Under new regulations defined in the 'Notice on the Prevention of Online Gaming Addiction in Juveniles', online video game providers became obligated to both prevent individuals under the age of 18 from playing for more than 1.5 hours each day (or 3 hours on a public holiday) and prevent these users from playing between the hours of 22:00 and 08:00 (ref. 6). These regulations were explicitly aimed to prevent the potential negative impacts of a heavy volume of video game consumption on physical and mental health among youth[48]. China's 2019 policy attracted substantial controversy, which only intensified after its expansion in September 2021 to limit minors to only a single hour of daily playtime between 20:00 and 21:00 on Fridays, Saturdays, Sundays and public holidays[46].

In addition to the debate on whether playtime impacts wellbeing, these restrictions have raised questions about their efficacy. Some have suggested that restricting playtime may simply lead to minors bypassing regulations, for example, by accessing games which do not or need not comply with regulations, using virtual private networks (VPNs)

**Table 1 | A matrix showing the OR of heavy play between each week preregulation and postregulation using dataset 1**

| | 0 | 1 | 2 | 3 | 4 | 5 | 6 | 7 | 8 | 9 | 10 |
|---|---|---|---|---|---|---|---|---|---|---|---|
| −1 | 0.96 | 0.94 | 0.95 | 0.91 | 0.99 | 0.95 | 1.03 | 1.01 | 1.15 | 1.22 | 1.56 |
| −2 | 0.89 | 0.88 | 0.88 | 0.84 | 0.92 | 0.89 | 0.96 | 0.94 | 1.07 | 1.13 | 1.46 |
| −3 | 1.03 | 1.01 | 1.02 | 0.97 | 1.06 | 1.02 | 1.10 | 1.09 | 1.24 | 1.31 | 1.68 |
| −4 | 0.78 | 0.77 | 0.77 | 0.74 | 0.80 | 0.77 | 0.84 | 0.82 | 0.94 | 0.99 | 1.27 |
| −5 | 0.71 | 0.70 | 0.70 | 0.67 | 0.73 | 0.71 | 0.76 | 0.75 | 0.86 | 0.90 | 1.16 |
| −6 | 1.00 | 0.99 | 0.99 | 0.95 | 1.03 | 1.00 | 1.08 | 1.06 | 1.21 | 1.27 | 1.64 |
| −7 | 0.77 | 0.75 | 0.76 | 0.72 | 0.79 | 0.76 | 0.82 | 0.81 | 0.92 | 0.97 | 1.25 |
| −8 | 0.79 | 0.78 | 0.78 | 0.75 | 0.82 | 0.79 | 0.85 | 0.84 | 0.96 | 1.01 | 1.29 |
| −9 | 0.76 | 0.74 | 0.75 | 0.72 | 0.78 | 0.75 | 0.81 | 0.80 | 0.91 | 0.96 | 1.23 |
| −10 | 0.68 | 0.67 | 0.67 | 0.64 | 0.70 | 0.67 | 0.73 | 0.71 | 0.81 | 0.86 | 1.10 |
| −11 | 0.65 | 0.64 | 0.64 | 0.61 | 0.67 | 0.64 | 0.69 | 0.68 | 0.78 | 0.82 | 1.05 |

Each column represents a different week postregulation; each row a different week preregulation. For example, the cell located at (−1,0) compares the odds of heavy play 0 weeks postregulation (that is the week spanning 1–6 November 2019) against the odds of heavy play 1 week preregulation (that is the week spanning 25–31 October 2019): its value (0.96) represents a situation in which individuals are 0.96 times as likely to engage in heavy play in week 0 when compared with week −1. All comparisons are significant at the $P < 0.001$ level using Fisher's exact test (two-sided).

that make an individual's location appear to not lie within Mainland China, or simply using an adult friend or relative's account[38,46,49]. Little is known about whether this is the case: under the 2019 regulations, game companies were individually tasked with monitoring playtime among their underage users, and restricting playtime accordingly, but their playtime data has not been made available to independent researchers[6]. A self-report data analysis suggested that South Korean playtime restrictions did not reduce playtime[50]. However, crucially, such data are not based on direct behavioural estimates of playtime, and therefore may be prone to error[18].

These questions reflect a broader set of concerns regarding the regulation of online behaviour and consumption among young people in general. To begin with, there are concerns that some specific policies may not work effectively and hence allow potentially harmful activities to continue unabated. For example, e-cigarette sales to minors are prohibited in all states within the USA, yet such products are still known to be widely purchased online by youth[51]. Additionally, there are concerns that other policies may 'backfire', and lead to the accidental emergence of novel sources of harm. For example, narratives around 'black market' gambling deal with the idea that overly stringent regulation of gambling may drive individuals towards unregulated, and potentially riskier, spaces such as cryptocurrency-based gambling sites[52–54].

In recent years, governments have sought to restrict minors' access to a variety of experiences and products online. These range from purchasing tobacco online to engaging in internet gambling and viewing pornography. Effective online regulation must overcome several key limitations. First, regulatory compliance must be consistently enforced among relevant corporate bodies in order to prevent displacement: the movement of individuals from compliant to uncompliant subsets of a sector[55]. For example, in Germany, the Interstate Treaty on the Protection of Minors in the Media requires pornographic websites to implement age verification. However, some websites have not complied with this order, leading to concerns that minors may continue to engage with these sites[56,57]. Similarly, while the online sale of cigarettes to minors is illegal in some parts of the USA, research has shown that a failure to implement age verification procedures has previously allowed the evasion of this law[58,59]. Research has also shown that cryptocurrency-based gambling providers frequently fail to implement identity verification for user registration, and allow individuals to deposit cryptocurrency for gambling without verification of the user's identity[53]. Indeed, in the video game domain itself, recent research has shown widespread industrial noncompliance with the legal requirement that companies remove gambling-like loot boxes from their games in Belgium[60].

Second, even if corporate compliance were universal, for the regulation of youth online behaviour to be effective, age verification checks must be non-trivial to evade[61]. Accurate and reliable identification of minors online is a challenging task[62]. The use of alternative identification documents and technological solutions provides a well-known avenue to regulatory escape in this domain[55,63]. Thus, despite attempts to age-restrict online access to products such as nicotine, gambling and pornography, their use by minors is common: in 2019, 11% of 11–16 year olds in the UK reported gambling during the prior week[64]. In 2021, a survey of adolescents found that the majority of respondents viewed online pornography, with many using a VPN[65]. In 2020, minors found themselves easily able to circumvent restrictions on the purchase of e-cigarettes by utilizing false identification[63].

When taken together, the evidence base suggests that the impact of Chinese regulations on playtime may be far from straightforward. It is crucial for behavioural science to provide evidence of the efficacy of such legislation. In this study, we therefore investigate the effectiveness of China's 2019 playtime regulation in reducing heavy play via two separate preregistered analyses of direct behavioural data: in the first of these, we analyse the prevalence of heavy play among more than 2.4 billion gamer profiles both before and after said regulations; in the second, we conduct a within-participants longitudinal analysis ($n = 10,000$) to determine whether individual gamers tended to play less heavily after restrictions were brought in.

## Results

### The odds of heavy gaming before and after regulation

On average 188,460,011 unique gamer profiles were seen in each of the 22 weeks of dataset 1, for a total of 2,486,192,234 unique profiles. The distribution of playtime between accounts and across weeks is depicted in Fig. 1.

An overall mean of 0.77% of gamer profiles engaged in heavy play before regulation and 0.88% after (dataset 1). Formal analysis of odds ratios (OR) using Fisher's exact test (two-sided) suggests that play tended to be significantly more heavy ($P < 0.001$) after regulation (OR = 1.14). However, this statistic does not reach our preregistered threshold for practical importance (OR = 2.00). A matrix showing the OR of heavy play between each week in our data is presented as Table 1; Fig. 2 shows the rate of heavy play for each week in our data (dataset 1).

**Sensitivity analyses.** We report below a variety of sensitivity analyses designed to check the robustness of our findings. These analyses were suggested during peer review and thus are not preregistered.

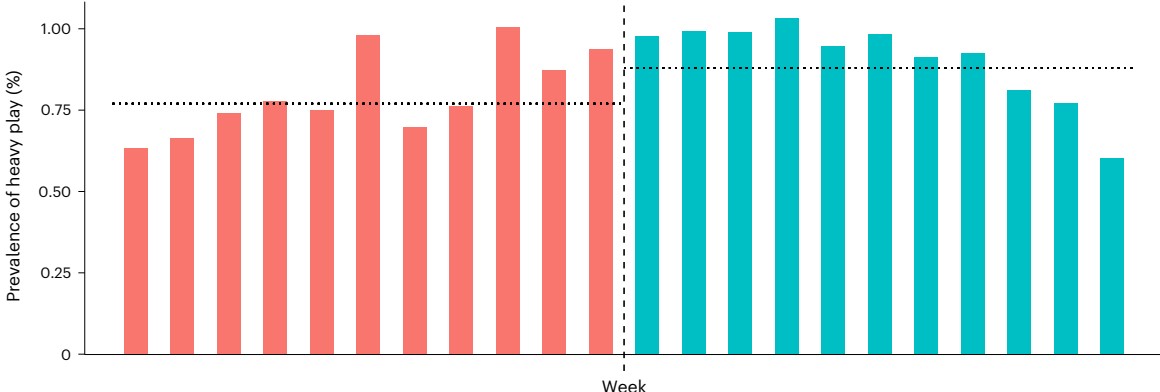

**Fig. 2 | The prevalence of heavy play for each of the 22 weeks in dataset 1.** Prevalence here represents the percentage of game profiles in each week's data that were playing for more than 4 hours per day, 6 days per week. The dashed line represents the implementation of regulations on 1 November 2019. Dotted horizontal lines represent the overall mean prevalence for both pre- and postregulation periods.

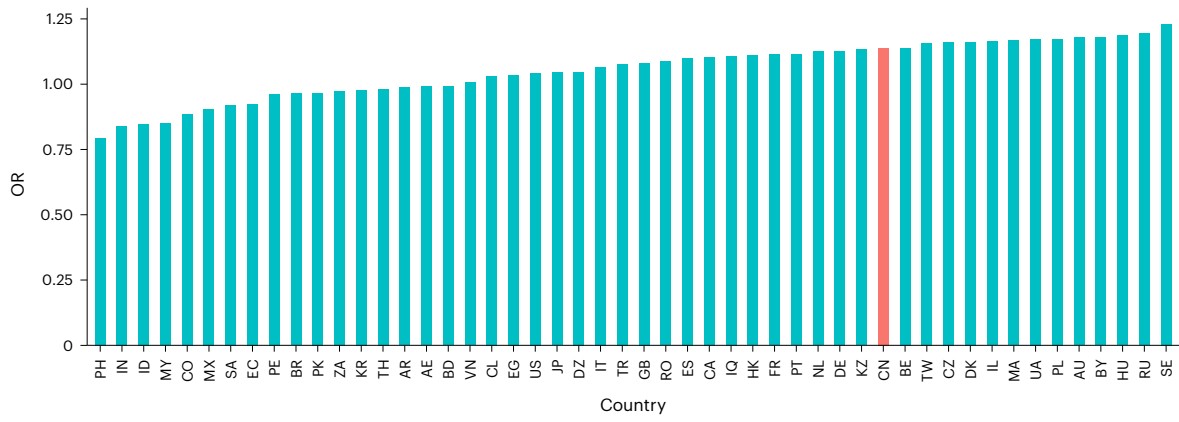

**Fig. 3 | OR for heavy play when comparing the 11 weeks before Chinese regulation with the 11 weeks following Chinese regulation (dataset 5).** The territories analysed here represent the 50 territories with highest average number of players in Unity's data for this period. China is highlighted in red (OR = 1.14). Countries are represented by their ISO2 country code.

First, we conducted 'difference-in-difference' analyses with a variety of global territories to examine whether the observed increases in heavy gameplay in our dataset were unique to China or exceptional. As noted above, play in China tended to be more likely to be heavy after regulation (OR = 1.14). However, during the same time period, similar or greater differences were observed in a variety of territories, ranging from Russia to Australia (Fig. 3).

Next, in order to more closely test any possible confounding effect of binarizing our outcome on our results, we treated playtime as a continuous variable. We examined whether the mean weekly playtime for a randomly selected account in a postregulation week still tended to be higher than a randomly selected account in a preregulation week. This was the case: after playtime, accounts numerically played for more hours each week. Before regulation, average playtime for any account during any given week was estimated at 1.64 hours (95% confidence interval (CI) 1.6404–1.6408). After regulation, it was estimated at 1.76 hours (95% CI 1.7582–1.7587). This suggests that the outcomes reported above are not well explained as a confounded product of our binary measurement scheme alone (Fig. 4). This was formally supported by the calculation of a partially overlapping *t*-test[66]. We compared both the mean probability that each account engaged in heavy play during the 11 weeks before regulation against the 11 weeks following regulation; and also the mean playtime for each account from the 11 weeks before regulation against the 11 weeks postregulation. Results suggested that not only did accounts tend to be more likely to play heavily postregulation (*t* = 102.942, d.f. = 2,321,091,203.249,

*P* < 0.001) but they also tended to log significantly more hours of play (*t* = 267.856, d.f. = 2,316,728,099.943, *P* < 0.001).

To be as conservative as possible, we then re-analysed both continuous and binary measures of playtime, but centring on 1 September 2021, at which time China's playtime mandates were adjusted to limit minors to only a single hour of daily playtime on Fridays, Saturdays, Sundays and public holidays[46]. Again, a similar lack of reduction in playtime was seen (Fig. 5). In the 11 weeks before these further adjustments, the odds of an individual account's weekly play being classified as heavy were estimated at 0.44% (95% CI 0.4423–0.004427); in the 11 weeks after adjustments, they were estimated at 0.59% (95% CI 0.5950–0.5957). Overall mean weekly playtime per account before adjustments was 1.29 hours (95% CI 1.2879–1.2882). Following adjustments, overall mean weekly playtime per account was estimated at 1.51 hours (95% CI 1.5066–1.5071). This was again formally supported by the calculation of a partially overlapping *t*-test[66]. We compared both the mean probability that an account engaged in heavy play during the 11 weeks before adjustments against the 11 weeks following adjustments; and also the mean playtime per account from the 11 weeks before adjustments against the 11 weeks after adjustments. We found that accounts not only tended to be more likely to play heavily postadjustments (*t* = 434.351, d.f. = 1,706,230,571.001, *P* < 0.001) but also tended to log significantly more hours of play (*t* = 680.604, d.f. = 1,693,333,300.856, *P* < 0.001).

Penultimately, we examined a reviewer suggestion to investigate the volume of account creations before and after regulation in

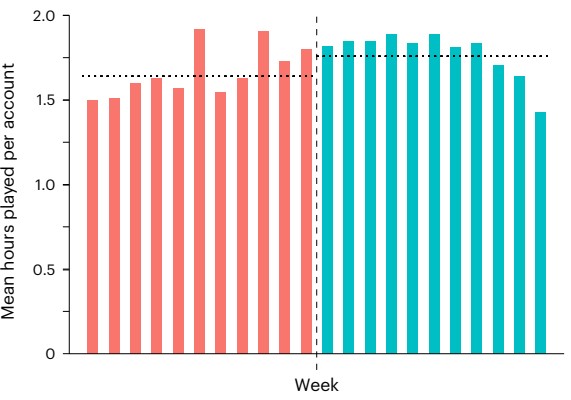

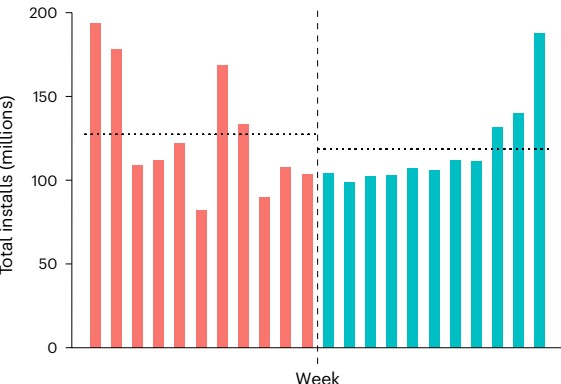

**Fig. 4 | Mean playtime (dataset 1) and total installs (dataset 4) for each of the 22 weeks preceding and following 1 November 2019.** Dashed lines represent the implementation of regulations on 1 November 2019. The chart on the left shows mean playtime for each week during this period. Dotted horizontal lines represent overall mean playtime for both pre- and postregulation periods. The chart on the right shows total installs for each of the 22 weeks. Dotted horizontal lines represent mean installs per week in China for pre- and postregulation periods (127.33 million and 118.57 million, respectively).

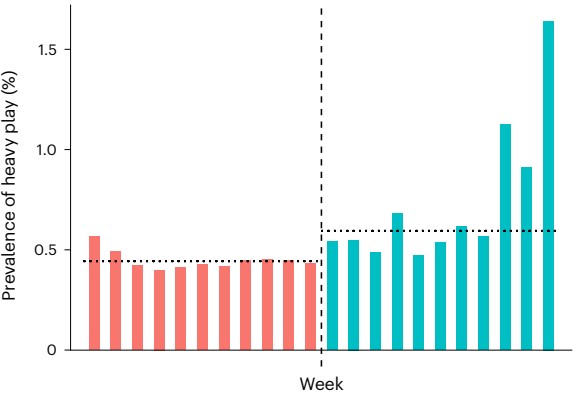

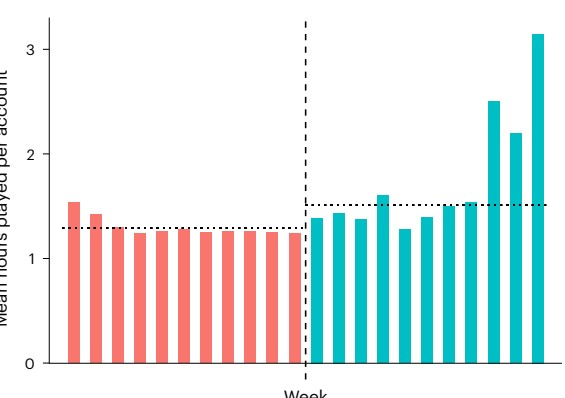

**Fig. 5 | Playtime for each of the 22 weeks in dataset 3.** The dashed line represents the implementation of regulations on 1 September 2021. The chart on the left shows the prevalence of heavy play for each week during this period, calculated in a consistent manner with the rest of our datasets (playing for more than 4 hours per day, 6 days per week). The chart on the right shows mean playtime per account for each week in this dataset. Dotted horizontal lines represent overall mean prevalence / playtime per account for both preregulation and postregulation periods. No reduction in either mean playtime or heaviness of playtime is observable postregulation.

order to indirectly assess whether youth might be creating additional accounts in response to regulation. We were unable to observe such a change (Fig. 4): a chi-squared goodness-of-fit test suggested that, if anything, fewer accounts were created during the 11 weeks following regulation when compared with the 11 weeks preceding the playtime mandates coming into effect (total installs before regulation were 1,400,722,598; total installs following regulation were 1,304,358,773, $\chi^2 = 3432794$, $P < 0.001$).

A final suggestion for sensitivity analysis was to reanalyse our data using a formal event study paradigm, as this is employed for causal inference in similar domains[67]. Event studies were conducted assessing the impact of regulations on individual accounts in China when compared with individual accounts from elsewhere in the East Asian cultural sphere. Difference in differences were assessed with outcomes represented both by increases in mean playtime per player and increases in the likelihood of heavy play. In all instances, analyses returned results that failed to show a statistically significant change in either mean playtime or the likelihood of heavy play after regulation among Chinese accounts when compared with accounts from elsewhere in the East Asian cultural sphere. For the purposes of brevity, these analyses and their results are reported in our supplemental materials rather than the main body of the paper.

## Differences in heavy play within individual gamers after regulation

We then focused on analysing changes to the likelihood of heavy play within individual gamers both before and after regulation, taking into account both the idea that individuals may have different propensities towards heavy play, and that general nonlinear trends may be seen in the heaviness of an individual's playtime. A multilevel logistic generalized additive model was fitted to the data (dataset 2). This model predicted whether an individual played heavily during any specific week in our data from a combination of (1) a nonlinear ('smooth') fixed effect of week; (2) a binary fixed effect representing whether regulations were in place during a week; and (3) random intercepts for each of the 10,000 gamer profiles in our dataset (dataset 2).

This model was able to explain 54.4% of variance in heavy playtime during the period in question. The smooth term associated with non-linear changes in the likelihood of heavy play was significant ($P < 0.001$, $\chi^2 = 199.1$, estimated degrees freedom = 5.68). Individuals tended to become less likely to play heavily with each passing week in an approximately linear fashion during the period under test (Fig. 6).

This unexpected effect led us to conduct a non-preregistered and exploratory sensitivity analysis. We built an identical model over an identical dataset of 10,000 individuals who played during the 22 weeks

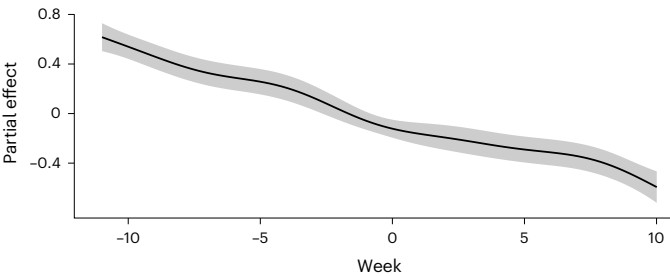

**Fig. 6 | Estimate of a nonlinear relationship of week on likelihood to play heavily.** Week 0 represents the first week after Chinese regulations came into play (that is the week commencing 1 November 2019). Overall, there appears to be an approximately linear decay in an individual's likelihood to engage in heavy gaming during the period under analysis. The shaded area represents a 95% CI. CIs are plotted using the default visualization package for big additive models within R's mgcv package and represent an area two standard errors above and below the smooth that was plotted.

surrounding 1 November 2017—a period where no restrictions were in place or introduced. Again, we found an approximately linear decline in the likelihood of heavy play within individuals, suggesting that this linear decline does not represent individuals somehow 'anticipating' regulation, but instead may be better explained by a phenomenon in which gamers tend to generally play a specific game less heavily the longer they have played that game for.

The binary variable associated with the presence/absence of Chinese playtime restrictions was also able to explain significant variance in the likelihood of heavy play (OR = 1.250, 95% CI of OR 1.078–1.448, $P = 0.003$). However, this effect was inverted in comparison to our prediction: individuals became more likely to play heavily following the implementation of restrictions. The smooth effect of weeks within the model is depicted in Fig. 6; QQ (quantile-quantile) and ACF (autocorrelation function) plots for the overall model are displayed in Fig. 7.

## Discussion

### No evidence for reduction in heavy gaming following playtime mandates

Heavy playtime is currently a topic of global concern, and state-mandated limits on playtime have currency as a potential policy lever for addressing this concern. However, previous research into their efficacy has been forced to rely on self-report evidence and has not analysed the impact of the most large-scale and far-reaching policy regarding playtime reduction: 'China's Notice on the Prevention of Online Gaming Addiction in Juveniles'.

To assess whether changes to heavy play occurred in China after this policy was implemented, we conducted two separate analyses using over seven billion hours' worth of playtime data. This data was exceptional both in the number of products it covered (over 1 million game identifiers) and the number of gamer profiles (over 2.4 billion). We found no evidence of reduced heavy playtime in our sample after China implemented its policy.

Initial analysis of the odds of any randomly selected Chinese player profile's weekly play being heavy found no practically significant difference in heavy play between any week before or after regulation, where practically significant was defined in our preregistration plan as an OR of 2.0: no weekly comparison reached or exceeded this value. Indeed, after 1 November, at a between-person level, individuals appeared 1.14 times more likely to engage in heavy play than before, and the overall prevalence of heavy gaming rose from 0.77% of weekly play to 0.88%. It is important to note that these levels fail to exceed our preregistered threshold for a meaningful effect and thus may be interpreted as consistent with methodological noise. This effect holds at the within-person level: after the restrictions, individual players were also

not more likely to play heavily postregulation. This data pattern undercuts contemporary debates regarding the societal impact of Chinese playtime regulation: regardless of whether one considers playtime limits a necessary public health measure or unwarranted state intrusion, our results suggest that the playtime restrictions implemented in China in 2019 are at a minimum not in the aggregate effective in reducing heavy playtime.

### Potential confounds

As noted above, our data suggest that heavy play in China did not decrease in prevalence following regulation. While such data are consistent with an ineffective policy, below we discuss a variety of alternative reasons why such a statistic may have been observed.

A first possible explanation is confounding due to history effects: if more public holidays occurred in the period under study after regulation than the period before regulation, heavy playtime due to public holidays may confound results. However, five more public holidays (six days: one for the Dragon Boat Festival and five for the National Golden Week) fell in the period before the restrictions than fell after the restrictions (one day: just New Year's Day), rendering this possible explanation inert. After a systematic search of relevant Chinese-language literature regarding holidays (Methods) we are aware of no other obvious history effects that may potentially confound results in such a manner.

A second possible explanation for the lack of reduction in heavy play observed here relates to a situation in which real reductions in playtime are confounded due to the binary classification scheme that we employ here. However, results of a sensitivity analysis comparing mean playtime for each week in our data were unable to provide evidence for such confounding: even when measured as a continuous variable, no reduction in playtime was observed following regulation (Fig. 4).

A final possible explanation is that true positive effects of regulation on minors are masked by majority adult players in our dataset. Chinese gaming company Tencent reports that only 6.4% of playtime in China on their games came from minors in September 2020[68]. It seems probable that a similar small fraction of individuals in our dataset were underage and thus subject to restrictions. In our data, playtime appeared heavier postregulation, albeit to a degree that did not meet our preregistered effect size threshold for practical importance (an OR of 2.0). However, it is crucial to note that we lack age information for each player in our dataset. This lack of metadata means that we cannot test whether inequal processes may be in operation simultaneously within the population under observation: for example, we cannot falsify the idea that an increase in heavy gaming among adults could be co-occurring with youth simultaneously playing less heavily. This lack of relevant demographic detail is a key limitation of the use of large-scale industry datasets such as the one employed here. We maintain that the result observed here is most plausibly explained by an ineffective policy. Nonetheless, in order to build on this work, future research must focus on generating data infrastructure: technological frameworks that allow privacy-preserving independent access to large-scale behavioural data fused to relevant self-report or demographic indicators[69,70].

### Lack of evidence for backfiring effect

One potential interpretation of the observed effect is that it represents a policy backfire: a situation in which Chinese policy was not simply ineffective, but in fact actively exacerbated the phenomenon it was attempting to suppress. We do not believe the data obtained here are consistent with such an explanation.

First, the effect observed here failed to exceed our preregistered threshold for a practically meaningful effect and thus is inconsistent with a policy backfire. It is important to note that a sceptic may suggest that our preregistered OR (2.0) may be overly conservative; that rigid cut-offs should not be followed unthinkingly; and that factors such as quality of measurement may influence the true size of a

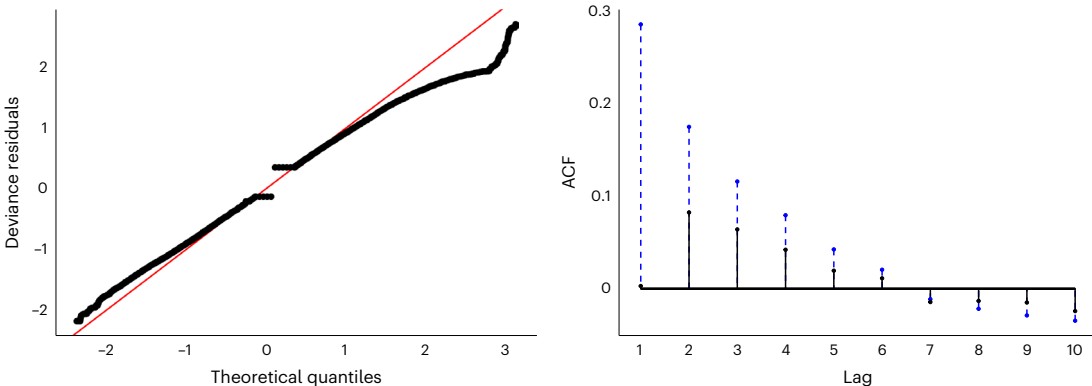

**Fig. 7 | QQ and ACF plots.** The left plot represents a QQ plot comparing deviance residuals with theoretical quantiles obtained by direct randomization of uniform quantiles. The right plot represents the ACF of both uncorrected and corrected models, with the reduction in autocorrelation associated with fitting AR(1) errors shown as a dashed line.

practically significant effect[71]. However, importantly, our sensitivity analyses reveal that the heaviness of playtime within any country may potentially be typified by a degree of volatility that is consistent with the OR observed for China in this case, rendering these arguments powerless. If one assumes that a country tends to play with completely stable levels of heaviness from one period to another, our observed OR between preregulation China and postregulation China may appear to reveal an important increase in heaviness of play. However, if countries routinely play more or less heavily to approximately this extent even in the absence of any formal regulation then this statistic provides weak support to such interpretations. Sensitivity analyses suggest that increases in heavy playtime of the sort observed in China during this period are routinely seen across the globe (Fig. 3). For example, countries as diverse as Belgium, Ukraine, Poland, Russia and Sweden all observed greater increases in heavy play during the postregulatory period than China did (all OR > 1.14). When viewed in this light, it is not appropriate to interpret the OR observed here as of practically meaningful magnitude.

**Plausible mechanisms for a lack of reduction in heavy play**
Why did we not find evidence of reductions in heavy play following regulation? There are several candidate explanations for this lack of evidence for a reduction in utilization.

One possible mechanism explaining this effect would be that pre-existing adult-associated player IDs tended to play more heavily postregulation, overshadowing reductions in heavy playtime among minor-associated IDs, because adults shared their account login details with minors postregulation (a known loophole of the regulation)[72]. In this scenario, a single adult-associated account used by several individuals may be involved in more hours of play per day while each individual using this account was continuing to play the same amount. This would account for both a lack of evidence for an increase in account creations and a lack of evidence for an increase in total playtime in China. However, it is important to note that we did not find evidence for an increase in account creation following regulation (Fig. 4). Indeed, fewer account creations (mean = 118.57 million) occurred in the weeks following regulation than in the weeks before regulation (mean = 127.33 million). Relatedly, individual players may have evaded restrictions by using a VPN[73]. Our within-participant sample intentionally included individuals who appeared to play in China before regulation, but could appear in any country postregulation, thereby allowing us to analyse individuals who may have been using VPNs: no reduction in heavy playtime was observable in this group either. Fully unpicking ID-sharing and VPN use as potential explanations is beyond the scope of this paper but must form a topic for future research.

A similar, further explanation would be inconsistent regulatory compliance across the games industry. Under the regulations, individual game providers are responsible for both ascertaining the real-life identity of each of their players; recording their age; and restricting their play accordingly. Very large stakeholders (such as the gaming company Tencent) have reported complying with this[68]. However, our data highlight the highly federated nature of the games industry: there are over one million game identifiers in our data, which are plausibly produced by tens of thousands of separate companies. A large portion of the global games industry consists of small 'independent' developers, which Unity Technologies is thought to primarily capture[74]. Top-down regulation may be able to secure compliance from large corporations who have the resources to effectively identify and police their player bases and have become prime targets of political intervention in China. It is less clear how compliance is easy to affect and police for thousands of small companies, particularly in light of similar noncompliance to top-down industry regulation in other parts of the globe. This uneven compliance may plausibly lead to either a lack of reduction in heavy playtime within small game companies, or even an increase, as heavy players migrate from now-regulated 'big' games by 'big', compliant companies to non-regulated 'small' games by non-compliant 'small' companies. We have seen similar phenomena in internet pornography regulation, where restriction of access to minors in one domain resulted in their displacement to unregulated spaces[55]. Our Unity data chiefly consists of small company games, and such a displacement migration may appear to be consistent with the increased likelihood of heavy gaming postregulation that we observed. However, we would suggest caution in this interpretation. It is crucial to point out that the observed OR in this study fell well below our preregistered threshold for practical importance.

Both possible explanations—player noncompliance and industry noncompliance—are broadly in line with prior policy literature on the regulation of online youth behaviour. It notes that effective regulation requires both widespread industry compliance and age verification mechanisms that are non-trivial for youth to overcome. As noted, our data cannot answer if either, both or neither pathways are responsible for the lack of reduction in utilization that we observed. Both are plausible. Youth throughout the globe share knowledge online about how to evade online age verification procedures[63–65]. Given the highly fragmented video game industry with a few outsized corporate actors surrounded by a majority of small and medium-sized enterprises, uneven industry noncompliance is similarly plausible. For tens of thousands of often extremely small and medium-sized enterprises, robust age verification procedures may be practically infeasible for companies to implement and for authorities to police.

## Conclusions

For the domain of gaming, our study provides evidence that broadly scoped restriction policies on youth digital behaviour may lead to no widespread and uniform decrease in utilization. While this is a notable contribution in and of itself, it is important to note two key limitations. First, our Unity dataset is pre-anonymized and contains no information about the age of the gamers in question. It will contain a mixture of both minor and adult gamers, only some of whom should be affected by China's playtime regulation. This means that while our analyses suggest that the likelihood of heavy playtime may have not been reduced in some parts of the games industry after regulation, they are unable to estimate how prevalent this phenomenon is among young people specifically. Second, and most important, our data present a view on only a specific portion of the market. Unity may be a common video game development engine, but not all games are made using Unity and their analytics solutions. Furthermore, Unity's comparative prominence as a game engine in China is unknown. It is possible that different patterns of engagement are seen in other contexts.

There is debate in the literature regarding whether China's top-down control of playtime is likely to promote the health and well-being of young people[49,73,75]. This paper suggests there might be a more fundamental underlying issue with such policies: they may be ineffective at causing intended changes to behaviour. This finding has important implications for the regulation of online gaming across the world. In analogous domains such as gambling, pornography and nicotine use, restriction of online youth access via mandatory bans has been associated with substantial regulatory escape. Here we show that a similar phenomenon may be occurring in the video game domain as well. This paper also forms a methodological blueprint for investigating how a broad range of regulatory measures may affect the technology sector. For example, future analyses could focus on the impact of the repeal of South Korea's restrictive Cinderella law, and the effects of attempted regulation of loot boxes in Belgium. Finally, this work highlights the utility of large-scale digital trace data as a methodological tool for understanding whether policy decisions in general lead to real-world behaviour change.

## Methods

### Preregistration

All analyses reported here were preregistered before both data analyses, and before data being downloaded from Unity's servers, unless stated otherwise. Preregistration took place on 24 September 2022 for analysis of dataset 1 and 17 July 2022 for analysis of dataset 2. Analyses of datasets 3–5 took place at the prompting of reviewers during the review process. These sensitivity analyses were not preregistered. Registration information for each of these analyses are available at the OSF repository associated with this project: https://doi.org/10.17605/OSF.IO/AUH2K.

### Datasets and preprocessing

The telemetry data used in this study span over one million separate game identifiers, 7.04 billion hours of playtime, and ~2.4 billion gamer profiles collected from Chinese users between 16 August 2019 and 16 January 2020. Access to telemetry data was provided by Unity Technologies, makers of the Unity engine, a development environment for games. Unity estimates that there are approximately five billion downloads of apps developed with Unity every month and that Unity is used by 61% of game developers[76]. The majority of games made with Unity are for mobile platforms. Games made in this engine commonly implement Unity Analytics, a play tracking service that allows developers to understand factors such as the daily playtime associated with individual users. Anonymized Unity Analytics telemetry from desktop and mobile games was the source of data for this study. It is important to note that our data were confined to this 22-week period in order to avoid bias and interference from the beginnings of the COVID-19

pandemic in China in early 2020: playtime has been shown to be heavily variable during the pandemic, with related containment and closure policies ('lockdowns') influencing a host of gaming-related variables[1].

Five separate datasets were used for this study.

Dataset 1: To investigate the odds of any individual player engaging in heavy gaming either before or after regulations, our individual-level data consisted of anonymized daily playtime logs for each individual gamer profile in the Unity dataset whose internet protocol (IP) address identified it as coming from Mainland China. Each log specified a unique identifier for an individual; a unique identifier for the game that individual had played on that day; and the period of time that individual had played for on that day. These data spanned 11 weeks before and after regulations were imposed on 1 November 2019 (that is 16 August 2019 to 16 January 2020, 22 weeks total). A sum total of 7.04 billion hours of playtime were recorded within this dataset, spread across ~2.4 billion (2,486,192,234) gamer profiles and ~1 million (1,175,923) game identifiers. This dataset is summarized in Fig. 1.

Preprocessing and extraction of this dataset from Unity's data lakes was a multi-step process. First, we filtered Unity's data so that our resulting dataset would not incorporate any playtime information for a product which was identified by Unity as not being a game. While the Unity engine is primarily intended for use in the games industry, it is possible to make non-game products using it. Unity indexes metadata for each product identified in its dataset. We filtered from our data any product whose metadata indicated that it was not a game. We then subsetted our data further so that we only obtained data which was identified as occurring within China. Every time an individual plays a game utilizing Unity Analytics, that playtime is tagged with the country associated with its IP. We thus subsetted our data to only include playtime which was associated with the ISO2 (International Organization for Standardization) country code 'CN'. Finally, we discretized our data into weekly chunks. For each individual gamer profile identifier in our filtered dataset, and for each week of our data, we calculated both the number of days that individual had played during that week, and the number of hours they had played for on each of those days. We used this data to create a binary 'heavy play' tag for each individual's weekly play (Measures). Weeks were defined as seven-day periods preceding or following 1 November 2019 (that is week 0 was 1–7 November 2019; week 1 was 8–15 November 2019; and so on).

Dataset 2: To investigate the impact of regulation on heavy gaming within individuals (that is, whether pre-existing gamers tended to reduce their heavy play after regulation), we utilized a smaller and more specialized individual-level dataset. The longitudinal nature of this analysis afforded us the ability to investigate heavy gaming among both individuals whose interactions consistently appeared to originate in China and those who appeared to play in China preregulation and other countries postregulation (that is, those who may have been using a VPN for such play). Thus, our base data for this study consisted of all individual gamer profiles in our data whose IP addresses identified them as being from China for every play session before 1 November 2019; and who engaged in play at least once during each of the 22 weeks under analysis here (that is 16 August 2019 to 16 January 2020). As shown in Fig. 1, the majority of individuals who interact with a game play that game only for a brief period of time. Our motivation for this cut-off was to avoid attempting to map longitudinal relationships across a dataset of individuals who played only during one week of the 22 weeks in question. As noted in our preregistration, we took only a subset of Unity's overall data to make analysis tractable via multilevel modelling: we randomly selected 10,000 individuals from this base to form our dataset (total playtime = 1,943,223 hours, total game identifiers = 1446). Tractability issues here related to our use of a multilevel logistic generalized additive model with gamer ID as a random effect: such a model was selected for its ability to model nonlinear changes in the heaviness of play over time (Analytical Approach). However, because such models involve the calculation of full penalty matrices

for each random effect, increasing the number of levels (that is gamers) causes nonlinear increases in the memory and time usage associated with model fitting[77]. For each of these individuals, our data recorded (a) the number of days that they played during each week under analysis; and (b) the total number of hours they played during each week under analysis. As noted below, these variables were transformed before analysis into a single binary indicator of whether play for a week was heavy.

Preprocessing and extraction from Unity's data lakes was again a multi-step process. First, we filtered Unity's data to remove non-game products as for dataset 1. Second, we filtered for location. Our procedure for this was subtly different to that for dataset 1: instead of taking only data that occurred within a CN location, we instead took data for individuals whose data always occurred within a CN location before regulation. Our motivation for this was to not necessarily exclude data from gamers using VPNs postregulation. We then filtered our data to only include individuals who played at least once during each of the 22 weeks under analysis. Our rationale for this was to avoid a scenario in which the majority of our longitudinal analysis took place over data from people who, for example, only picked up and played a game once: such an analysis would not constitute a severe test of our hypotheses. When we had established these filtering criteria, we took a random sample of 10,000 gamer profiles, and calculated heavy play for each of the 22 weeks of these profiles' data in an identical manner to the one employed for dataset 1.

Overall, gamers in this dataset played a median 5.55 hours per week (first quartile: 2.78, third quartile: 10.39), spread across a median of 6.14 (first quartile: 5.09, third quartile: 6.77) days. Within our sample, 2431 gamer profiles (24.31%) engaged in heavy play on at least one occasion (one week) during the study: 16,613 of the 220,000 weekly play summaries recorded involved heavy play, sensu the definition used here (7.55% of sample).

Dataset 3: Dataset 3 was generated for the purposes of the sensitivity analyses that were conducted following peer review. Data were filtered in an identical manner to dataset 1. Variables were calculated in an identical manner to dataset 1. The sole difference between dataset 1 and dataset 3 is that dataset 3 aggregates data drawn from the 11 weeks preceding and following 1 September 2021 (when China's regulation was further adjusted), rather than the 11 weeks preceding and following 1 November 2019 (when China's 'Notice on the Prevention of Online Gaming Addiction in Juveniles' first came into effect). A sum total of 6.53 billion hours of playtime were contained within this dataset.

Dataset 4: Dataset 4 was generated for the purposes of the sensitivity analyses that were conducted following peer review. Data were filtered in an identical manner to dataset 1. However, rather than measuring playtime per week, dataset 4 instead measures account creations per week. Every time a user accesses a new game (that is a game that is new to their device), Unity records this account creation in their data lakes. We calculated the sum total installs for each of the 22 weeks immediately preceding and following China's regulation on 1 November 2019. Weeks were again defined as seven-day periods preceding or following 1 November 2019 (that is week 0 was 1–7 November 2019; week 1 was 8–15 November 2019; and so on). In total, this dataset records 2.70 billion account creations during this period.

Dataset 5: Dataset 5 was created for the purpose of sensitivity analysis following peer review. It may be thought of as a superset of dataset 1. It is identical in every way to dataset 1, except it contains separate data for the 50 territories with highest average number of players per week in Unity's data for this period, rather than just China. These territories are as follows: Algeria, Argentina, Australia, Bangladesh, Belarus, Belgium, Brazil, Canada, Chile, China, Colombia, Czechia, Denmark, Ecuador, Egypt, France, Germany, Great Britain, Hong Kong, Hungary, India, Indonesia, Iraq, Israel, Italy, Japan, Kazakhstan, Malaysia, Mexico, Morocco, the Netherlands, Pakistan, Peru, the Philippines,

Poland, Portugal, Romania, Russia, Saudi Arabia, South Africa, South Korea, Spain, Sweden, Taiwan, Thailand, Turkey, UAE, Ukraine, USA, Venezuela.

**Data limitations.** The work presented here relies on analysis of data from video games created with, or using components from, the Unity game engine, where Unity Analytics is enabled. Additionally, the user accounts in the data under analysis are not linked between individual products made using Unity Analytics: if one individual played two separate games, they would appear as two separate individual gamer profiles. Thus, our data are unable to model phenomena in which individuals cycle between multiple separate games. Given that our dataset contains over two billion gamer profiles, and demographic estimates currently place the total population of China below this figure, it is likely that this phenomenon is widespread in our data.

However, the most important limitation of the data utilized here is its lack of personally identifiable information: as described below, we are unable to identify individual players, or any features of those players beyond their broad geographical location. Thus, our dataset will naturally incorporate both individuals aged under 18 (whose play is theoretically regulated) and those aged 18 or above (whose play is not regulated). This inability to disentangle relevant from irrelevant gamers is key to the analytic strategy employed here.

Finally, as noted in our literature review, China's 2019 playtime mandate is characterized by permitting different hours of play on days that are public holidays and those that are not. On public holidays, minors may play for 3 hours per day—double the 1.5 hours that is permissible on other days. This presents a potential confound: if public holidays occur more frequently in the period after regulation, they may artificially inflate playtime for this period, leading researchers to inappropriately conclude either that the policy was ineffective or that it may have even increased the prevalence of heavy play. In order to investigate the potentially confounding effects of public holidays, a member of the research team who is fluent in Chinese examined official government sources regarding public holidays during this period[78,79]. During the period under analysis, the following dates were designated as public holidays: 13 September 2019 (Dragon Boat Festival); 1–7 October 2019 (National Golden Week); 1 January 2020 (New Year's Day). Of these, only 1 January 2020 falls after regulation. In addition to this, two specific weekend dates before regulation (29 September, 12 October) were used as workdays to allow for the National Golden Week of uninterrupted holidays. Thus, there were six public holiday days before regulation and one following regulation.

## Measures

This manuscript focuses on measures of heavy gaming. As preregistered, we exclusively measure heavy gaming via the formulation suggested by Colder-Carras et al.[31], who suggest a plausible threshold as an individual playing for 4 or more hours per day, 6 or more days per week[31]. We chose this criterion among alternatives because it is comparatively conservative (and thus a more defensible and severe test of regulatory effectiveness), and empirically informed, based on prior qualitative and population-level studies.

For each individual in our dataset, we calculate whether their gaming was heavy during that week as a binary variable and use this as our measure of play. This is a deliberate measurement strategy specified in our preregistration: our aim here was not to determine whether there was a change in, for example, the average number of hours played by each individual after regulation occurred, but to measure the more conceptually aligned variable of whether heavy play became less common after regulation.

## Analytical approach

In this manuscript, we approach each of our individual datasets via different analytic strategies.

**The odds of heavy play before and after regulation.** Our first analysis takes place over dataset 1 and aims to investigate whether heavy playtime became less common in China after regulation came into effect in November 2019. To undertake this analysis, we follow a relatively simple but interpretable analytic strategy: we calculate the OR between each of the 11 weeks preregulation against each of the 11 weeks postregulation, forming a matrix of 121 comparisons. In a deviation from our preregistered plan, we additionally compute an overall comparison representing the odds that a randomly selected player from one of the 11 weeks preregulation is playing heavily versus a randomly selected player from one of the 11 weeks postregulation. This was undertaken to allow comparisons between these overall periods to be understood more easily: we were concerned that requiring requiring readers to individually interpret the effect of regulation on each separate week in our data might prove ineffective or confusing. As our sample sizes are very large here, statistical significance does not provide a stringent enough benchmark for establishing the existence of a practically meaningful effect.

We preregistered an OR of 2.0 as a minimal practically important difference, as suggested in Ferguson's recommendations for practical significance in social science research[71]. In concrete terms, this would represent a scenario in which individuals were more than two times as likely to play heavily in any preregulation week compared with any postregulation week.

**The reduction of heavy playtime within gamers after regulation.** Our second analysis takes place within participants, and aims to establish whether individual Chinese gamers were likely to play less heavily after regulations came into force on 1 November 2019. In concrete terms, this model aims to estimate whether an individual is less likely to engage in heavy gaming after regulation, when taking into account both individual differences in tendency towards heavy gaming, and also any nonlinear trends in heavy gaming during this period (for example, the longer an individual plays a game, the less likely they may be to play heavily).

To estimate this, we fit a multilevel logistic generalized additive model to our data, with heaviness of gaming in a specific week as a binary outcome (Measures). Fixed predictors consist of both the binary impact of regulations, and a smooth term representing any overall nonlinear trend in heavy gaming within our sample separate to this. In order to account for individual differences in heavy playtime, we also include random intercepts for each gamer profile in our dataset. As preregistered, to account for temporal dependency in model residuals, we correct for potential autocorrelation in these residuals[80,81]. We did this by first fitting an uncorrected model; estimating an autocorrelative process in model residuals using Hyndman–Khandakar's procedure; and then refitting an updated model[82]. In deviation from our preregistered plan, we were only able to consider AR(1) processes (i.e., those modelled using an autoregressive model of order 1) rather than ARMA processes (i.e., those modelled using autoregressive moving average models of arbitrary order) in model residuals due to issues with computational tractability. The reason for this deviation was practical in nature. The large number of random intercepts in our model necessitated the use of specialized generalized additive modelling tools for very large datasets (that is, the BAM function in R's mgcv package). This approach does not afford the ability to adjust for multi-parameter structures in a model's errors, necessitating this deviation.

**Ethics and inclusion.** This project involved the analysis of data ultimately collected within the People's Republic of China. To ensure that the research conducted here was locally relevant and accurate, our research team incorporated a member of the community under analysis here: a Chinese national with substantial knowledge regarding local regulation. At the time of authoring this manuscript this individual was affiliated with a European university in their role as a research student. This individual was involved in the holistic research process from initial study design through to authoring the manuscript. They were not involved in negotiating data ownership in this case. Roles and responsibilities were determined collaboratively ahead of the research process. No capacity-building programmes for local researchers were discussed in this case.

Ethical approval for this study was given by the Physical Sciences Ethics Committee at the lead author's host institution: the University of York (application identifier: Zendle20211021). The data used in the current study does not include personally identifiable information and the research team does not have access to personally identifiable information from Unity Analytics. The data used in the study is pseudonymized by way of a token unique to each player of each individual game—no players are traceable across games, nor are players identifiable from this data.

Unity Technologies collect and store user-generated data using a plug-in for its engine known as Unity Analytics. Games that make use of this package are required to incorporate a consent agreement when users engage with the game in question, which explains the collection and use of this data to the player. Unity Technologies has public documentation that explains the requirements for collection, storage and use of analytics data to developers, available on the company's website. Included in the list of uses of the collected data, explained in the collection agreement, is research purposes, which is the purpose under which this data has been shared with the current research team. Collectively, the ethics committee that reviewed our research protocol determined this use of the data to be compliant with research ethics norms and expectations.

**Reporting summary**

Further information on research design is available in the Nature Portfolio Reporting Summary linked to this article.

## Data availability

Data for this study were provided by Unity Technologies under a data-sharing agreement between this organization and the lead author's host institution. The data under analysis here cannot be made publicly available. Other researchers interested in data access must contact Unity Technologies themselves. This restriction on public access is grounded in the terms of the data-sharing agreement formed between Unity Technologies and the lead author's host institution and is linked to the commercially sensitive nature of these large-scale data. No fee was paid to Unity Technologies for accessing this data. Usage fees were paid to Google for the transfer of data using the Google BigQuery serverless data warehouse.

## Code availability

Data were extracted from Unity databases using Google BigQuery. Data analysis was conducted using R (v.4.0.2). Our pipeline involved first downloading data from Google BigQuery using Structured Query Language-like code. The BigQuery code that supports the download and aggregation of data incorporates information regarding the proprietary structure of Unity's data lakes and is therefore not publicly available. The R code is available at the OSF repository associated with this project: https://doi.org/10.17605/OSF.IO/AUH2K

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

## Acknowledgements

D.Z., A.D. and C.F. have no relevant funding to declare for this project. Data access was funded via an annual stipend that D.Z. receives from his host institution (the University of York). N.B. and E.G.P. are supported by the EPSRC Centre for Doctoral Training in Intelligent Games and Game Intelligence (IGGI), grant code EP/S022325/1. L.X. is supported by a PhD Fellowship funded by the IT University of Copenhagen (IT-Universitetet i København), which is publicly funded by the Kingdom of Denmark (Kongeriget Danmark). The funders had no role in study design, data collection and analysis, decision to publish or preparation of the manuscript. The authors wish to acknowledge the support provided by M. Underwood and Unity Technologies in providing the data access which underpins this work.

## Author contributions

All authors jointly conceived and designed the study. L.X. provided specific guidance regarding the legal context in the People's Republic of China. N.B. and A.D. provided specific guidance regarding statistical analysis. D.Z. analysed the data. All authors jointly wrote the paper. D.Z. was responsible for redrafting the paper. Data access was negotiated by A.D. and D.Z.

## Competing interests

Data for this study were provided by Unity Technologies. Unity Technologies played no role in the design of the study, its reporting or its execution. No funds were disbursed to the research team for the work incorporated into this manuscript. This includes data access fees for the serverless data warehouse from which the data under analysis were accessed, which were paid by the lead author out of a research stipend he receives from his host institution. Access to the data used in this study was contingent on a data-sharing agreement between Unity Technologies and D.Z.'s host institution. Legal approval was sought from Unity for the sharing of the data in this paper before its submission for peer review. D.Z. has never received any form of funding from the games industry. He has worked as a paid consultant for governments seeking to understand the effects of video games and gambling. He has worked as an expert witness in cases relating to the video game industry, but has never represented the games industry legally or been formally affiliated with any games industry body in any way. D.Z. has been involved in brokering data-sharing agreements with industry stakeholders in the past. He acknowledges that such data-sharing agreements constitute a conflict of interest as important as financial awards and wishes to highlight that he has used such data brokerage in ways that are likely to give him indirect financial advantage: he has used them as evidence for excellence in promotion applications; he has used them as evidence in grant applications. No such applications have been funded at the time of writing this manuscript. D.Z. is a member of the Advisory Board for Safer Gambling, a statutory body whose remit is to provide independent advice to the UK Gambling Commission. D.Z. is the recipient of an Academic Forum for the Study of Gambling Major Exploratory Grant that is derived from 'regulatory settlements applied for socially responsible purposes' received by the UK Gambling Commission and administered by Gambling Research Exchange Ontario (GREO). He has no further conflicts to declare. C.F. has received funding from the Tides Foundation on the recommendation of the Unity Charitable Fund (grant number TF2201-105180) for a separate project. A.D. has previously worked in a paid capacity within the video games industry as a game analytics consultant. He has worked on multiple industry-focused projects with a focus on knowledge transfer. He has received funding from the Tides Foundation on the recommendation of the Unity Charitable Fund (grant number TF2201-105180) for a separate project. L.Y.X. was employed by LiveMe, then a subsidiary of Cheetah Mobile (NYSE:CMCM), as an in-house counsel intern from July 2019 to August 2019 in Beijing, People's Republic of China. L.Y.X. was not involved with the monetisation of video games by Cheetah Mobile or its subsidiaries. L.Y.X. undertook a brief period of voluntary work experience at Wiggin LLP (Solicitors Regulation Authority (SRA) number: 420659) in London, England in August 2022. L.Y.X. has contributed and continues to contribute to research projects that were enabled by data access provided by the video game industry, specifically Unity Technologies (NYSE:U) (October 2022 to Present). L.Y.X. was the recipient of two AFSG (Academic Forum for the Study of Gambling) Postgraduate Research Support Grants that were derived from 'regulatory settlements applied for socially responsible purposes' received by the UK Gambling Commission and administered by Gambling Research Exchange Ontario (GREO) (March 2022 & January 2023). L.Y.X. has accepted funding to publish academic papers open access from GREO that was received by the UK Gambling Commission as above (October, November, & December 2022). L.Y.X. has accepted conference travel and attendance grants from, inter alia, the organisers of the 13th Nordic SNSUS (Stiftelsen Nordiska Sällskapet för Upplysning om Spelberoende; the Nordic Society Foundation for Information about Problem Gambling) Conference, which received gambling industry sponsorship (January 2023). L.Y.X. has received an honorarium from the Center for Ludomani for contributing a parent guide about a mobile game for Tjekspillet.dk, which is funded by the Danish Ministry of Health's gambling addiction pool (Sundhedsministeriets Ludomanipulje) (March 2023). The up to date version of L.Y.X.'s conflict of interest statement is available via https://sites.google.com/view/leon-xiao/about/conflict-of-interest. EP has previously received funding from the Academic Forum for the Study of Gambling (AFSG) Postgraduate Research Support Grant that was derived from 'regulatory settlements applied for socially responsible purposes' received by the UK Gambling Commission and administered by Gambling Research Exchange Ontario (GREO). The remaining authors declare no competing interests.

## Additional information

**Correspondence and requests for materials** should be addressed to David Zendle.

# Reporting Summary

## Statistics

For all statistical analyses, confirm that the following items are present in the figure legend, table legend, main text, or Methods section.

| n/a | Confirmed | |
|---|---|---|
| ☐ | ☒ | The exact sample size (*n*) for each experimental group/condition, given as a discrete number and unit of measurement |
| ☐ | ☒ | A statement on whether measurements were taken from distinct samples or whether the same sample was measured repeatedly |
| ☐ | ☒ | The statistical test(s) used AND whether they are one- or two-sided *Only common tests should be described solely by name; describe more complex techniques in the Methods section.* |
| ☐ | ☒ | A description of all covariates tested |
| ☐ | ☒ | A description of any assumptions or corrections, such as tests of normality and adjustment for multiple comparisons |
| ☐ | ☒ | A full description of the statistical parameters including central tendency (e.g. means) or other basic estimates (e.g. regression coefficient) AND variation (e.g. standard deviation) or associated estimates of uncertainty (e.g. confidence intervals) |
| ☐ | ☒ | For null hypothesis testing, the test statistic (e.g. *F*, *t*, *r*) with confidence intervals, effect sizes, degrees of freedom and *P* value noted *Give P values as exact values whenever suitable.* |
| ☒ | ☐ | For Bayesian analysis, information on the choice of priors and Markov chain Monte Carlo settings |
| ☒ | ☐ | For hierarchical and complex designs, identification of the appropriate level for tests and full reporting of outcomes |
| ☐ | ☒ | Estimates of effect sizes (e.g. Cohen's *d*, Pearson's *r*), indicating how they were calculated |

*Our web collection on statistics for biologists contains articles on many of the points above.*

## Software and code

Policy information about availability of computer code

| Data collection | Google BigQuery (no relevant versioning that we are aware of) was used to collect the data in this study. This is a cloud-based serverless data warehouse with an associated SQL-like query language. |
|---|---|
| Data analysis | Data analysis was conducted using R (v.4.0.2). An OSF repository containing relevant code is available at: doi.org/10.17605/OSF.IO/AUH2K |

For manuscripts utilizing custom algorithms or software that are central to the research but not yet described in published literature, software must be made available to editors and reviewers. We strongly encourage code deposition in a community repository (e.g. GitHub). See the Nature Portfolio guidelines for submitting code & software for further information.

## Data

Policy information about availability of data

All manuscripts must include a data availability statement. This statement should provide the following information, where applicable:

- Accession codes, unique identifiers, or web links for publicly available datasets
- A description of any restrictions on data availability
- For clinical datasets or third party data, please ensure that the statement adheres to our policy

Data for this study were provided by Unity Technologies under a data sharing agreement between this organization and the lead author's host institution (the University of York). The data under analysis here cannot be made available to external scholars. This restriction on public access is grounded in the terms of the data

sharing agreement formed between Unity Technologies and the lead author's host institution and is linked to the commercially sensitive nature of these large-scale data.

## Research involving human participants, their data, or biological material

Policy information about studies with human participants or human data. See also policy information about sex, gender (identity/presentation), and sexual orientation and race, ethnicity and racism.

| | |
|---|---|
| Reporting on sex and gender | There were no participants in this research - it was conducted purely on anonymised secondary data. |
| Reporting on race, ethnicity, or other socially relevant groupings | There were no participants in this research - it was conducted purely on anonymised secondary data. |
| Population characteristics | There were no participants in this research - it was conducted purely on anonymised secondary data. |
| Recruitment | There were no participants in this research - it was conducted purely on anonymised secondary data. |
| Ethics oversight | This study protocol was approved by the Physical Sciences Ethics Committee at the University of York. |

Note that full information on the approval of the study protocol must also be provided in the manuscript.

## Field-specific reporting

Please select the one below that is the best fit for your research. If you are not sure, read the appropriate sections before making your selection.

☐ Life sciences   ☒ Behavioural & social sciences   ☐ Ecological, evolutionary & environmental sciences

For a reference copy of the document with all sections, see nature.com/documents/nr-reporting-summary-flat.pdf

## Behavioural & social sciences study design

All studies must disclose on these points even when the disclosure is negative.

| | |
|---|---|
| Study description | This is a quantitative observational study, involving longitudinal measurements at both population and within-participant levels. In brief, in November 2019 the People's Republic of China (PRC) enacted a novel policy which was designed to limit the amount of playtime that individuals were able to engage in within video games. In this study, we test the impact of this policy on heavy play.<br><br>We test this via two primary preregistered analyses: In a first analysis, we investigate whether the overall prevalence of heavy play decreases week-on-week after regulation is put in place.<br><br>In a second, within-participant, analysis, we take a sample of accounts who were active both before and after regulation, and investigate whether the presence of playtime regulations can explain substantial variance in the likelihood of these accounts playing heavily on any given week.<br><br>Several further sensitivity analyses are undertaken, with unique datasets underpinning these. These data range from counts of the number of video game installs in PRC during the period in question; to sensitivity analyses over the number of hours played during the period in question (Rather than the prevalence of heavy play); to an analysis of an analogous period during 2021. |
| Research sample | The primary sample under analysis are 2,486,192,234 unique profiles of gamers drawn from games that utilise Unity Analytics, and which were geolocated in PRC during the 22 week period under study. This sample is unable to represent the entire games market in PRC (many games are not developed using Unity, and patterns of play may vary in these games). However, it is able to represent the population of gamers who engage with Unity products, which is substantial (over 7bn hours of playtime within this dataset alone) and of real-world importance.<br><br>For the second, within-participants analysis, we subsampled 10,000 accounts from within this dataset which were active during all 22 weeks under study. This data is thus able to represent the subset of players who played games made using Unity Analytics, and who played consistently during this period within the PRC.<br><br>The source for all datasets and analyses described above were Unity Technologies' internal data lakes, which anonymously record the presence of play sessions in games which implement Unity Analytics. |
| Sampling strategy | Data in our primary dataset are not subsampled at all: They represent every account engaging with a product that implements Unity Analytics within the PRC during the period of interest.<br><br>For our within-participants analysis, our base data consisted of all individual gamer profiles in our data whose IP addresses identified them as being from China for every play session prior to 1st November 2019; and who engaged in play at least once during each of the 22 weeks under analysis here (i.e. 16th August 2019 to 16th January 2020). We randomly selected 10,000 accounts from this base to form our dataset. The reason for the subsampling was computational tractability: The modelling approach selected for this study would have taken an infeasibly large amount of memory and time if the sample was significantly larger. |

| | |
|---|---|
| Data collection | The data used in this study is data from Unity Analytics that is pseudonymised by way of a token unique to each player of each individual game - no players are traceable across games, nor are players identifiable from this data. Unity technologies collect and store user-generated data using a plug-in for its engine known as Unity Analytics. The researcher was not blinded to experimental conditions and/or study hypotheses when collecting this data. |
| Timing | The primary data are drawn from a 22-week period surrounding Nov 1st 2019 - the 11 weeks (i.e. 7-day periods) immerdiately preceding and following this date. |
| Data exclusions | The subset of Unity data used for this study was generated by selecting all daily playtime logs which (a) were identified as taking place within the PRC; (b) took place on a date equal to or greater than 16th August 2019; (c) Took place on a date equal to or less than 16th January 2020. A final exclusion criterion was that all products which implemented Unity Analytics but whose metadata identified them as not being a game were not included in this study. |
| Non-participation | No participants were involved in this study. |
| Randomization | This is an observational study taking place over historic secondary data, and thus typical randomisation methods (e.g. assigning participants to specific conditions) are not applicable here. The only case in which randomisation could be said to be employed during this study was the selection of 10,000 random accounts for the analysis of within-person effects, as outlined above. |

# Reporting for specific materials, systems and methods

We require information from authors about some types of materials, experimental systems and methods used in many studies. Here, indicate whether each material, system or method listed is relevant to your study. If you are not sure if a list item applies to your research, read the appropriate section before selecting a response.

## Materials & experimental systems

| n/a | Involved in the study |
|---|---|
| ☒ | ☐ Antibodies |
| ☒ | ☐ Eukaryotic cell lines |
| ☒ | ☐ Palaeontology and archaeology |
| ☒ | ☐ Animals and other organisms |
| ☒ | ☐ Clinical data |
| ☒ | ☐ Dual use research of concern |
| ☒ | ☐ Plants |

## Methods

| n/a | Involved in the study |
|---|---|
| ☒ | ☐ ChIP-seq |
| ☒ | ☐ Flow cytometry |
| ☒ | ☐ MRI-based neuroimaging |

