## [Peer Review File · Nature Human Behaviour]

Peer Review Information

Journal: Nature Human Behaviour

Manuscript Title: No evidence that Chinese playtime mandates reduced heavy gaming in one segment of the video games industry

Corresponding author name(s): David Zendle

Reviewer Comments & Decisions:

Decision Letter, initial version:

9th January 2023

Dear Dr Zendle,

Thank you once again for your manuscript, entitled "Chinese playtime mandates paradoxically led to heavier gaming in one segment of the video games industry", and for your patience during the peer review process.

Your Article has now been evaluated by 3 referees. You will see from their comments copied below that, although they find your work of potential interest, they have raised quite substantial concerns. In light of these comments, we cannot accept the manuscript for publication, but would be interested in considering a revised version if you are willing and able to fully address reviewer and editorial concerns.

We hope you will find the referees' comments useful as you decide how to proceed. If you wish to submit a substantially revised manuscript, please bear in mind that we will be reluctant to approach the referees again in the absence of major revisions. We are committed to providing a fair and constructive peer-review process. Do not hesitate to contact us if there are specific requests from the reviewers that you believe are technically impossible or unlikely to yield a meaningful outcome.

In your revision, we ask that you address all reviewers' comments. In particular, please follow Reviewer #1's suggestion to strengthen your causal inference by, if data allows, implementing difference-in-differences analyses. Please also follow reviewers' advice in implementing age-based analyses, again if your data allows for this, and rule out the possible effect of holidays. Additionally, we ask that you do not interpret your results as showing a backfiring effect, but rather the lack of an effect, and frame the narrative around spillover effects rather than direct effects on adolescents if your data does not allow you do stratify by age.

If you wish to submit a suitably revised manuscript we would hope to receive it within 4 months. I would be grateful if you could contact us as soon as possible if you foresee difficulties with meeting this target resubmission date.

- Include a "Response to the editors and reviewers" document detailing, point-by-point, how you addressed each editor and referee comment. If no action was taken to address a point, you must provide a compelling argument. When formatting this document, please respond to each reviewer comment individually, including the full text of the reviewer comment verbatim followed by your response to the individual point. This response will be used by the editors to evaluate your revision and sent back to the reviewers along with the revised manuscript.
- Highlight all changes made to your manuscript or provide us with a version that tracks changes.

[REDACTED]

Thank you for the opportunity to review your work. Please do not hesitate to contact me if you have any questions or would like to discuss the required revisions further.

Sincerely,

Arunas Radzvilavicius, PhD
Editor, Nature Human Behaviour
Nature Research

Reviewer expertise:

Reviewer #1: psychology and behavioural science of gaming and gambling, exp

Reviewer #2: psychology of gaming

Reviewer #3: economics

REVIEWER COMMENTS:

Reviewer #1:

Remarks to the Author:

This manuscript looks at Chinese video game play before and after a policy restricting youth use, and finds a paradoxical increase in total play after the policy came into place. The manuscript is well-motivated via discussion of issues in (youth) consumer protection in similar risky domains, and is an inventive use of mass real world data. My comments are as follows

p2 lines 70-85 There is discussion here of in my view two separate cases. You can have policies that simply do not work effectively and which allow risky behaviour to continue (e.g., US cigarette limitations). Then you have a policy that backfires by driving users toward more risky content (e.g., restrictions on conventional online gambling leading to users going to crypto gambling sites). I believe this second case is more relevant to this work here. The pornography example given on line 224 in the Discussion is a nice example of a policy backfiring, as the present results appear to show, and I think that would be a better motivator than e.g. the loot boxes example given here (which is an example of a policy merely being ineffective).

p2 line 91 The thing is, if you dig into the UK child gambling stats then most children gambling are either doing so legally or with parental approval. For example, in this wave the national lottery could be used legally by 16 year olds (the law changed in 2020), and also private gambling between friends is legal for children to do. Instances when a child does gamble online are rare, and are overwhelmingly cases where they are doing so with explicit parental approval. This does make me think that parents giving their IDs to their children in order to continue gaming may well be a driver of these effects. The authors mention this a little but do not provide a compelling case about why this is not driving the effect. Overall, I think part of the observed effect might be due to the binarization of the outcome (gamers categorised either as heavy gamers or not). Some children might play exactly the same amount before/after the policy, but if they tend to start spreading their play across multiple IDs AFTER the policy, then this may make it look like there is a greater number of heavy gamers in the population as defined in this binary sense by the authors. Some analysis treating heavy play as a continuous variable would I think be needed to counter this argument.

There is some discussion of other potential drivers of the effect (e.g., national holidays), but I think this type of econometric-type analysis should go further to increase confidence in the results. For example, the Authors could compare against a culturally-similar country which did not see the policy change. Could a difference-in-difference analysis be conducted with for example Taiwan as the control case in order to better convince that the policy change is driving this overall trend?

Reviewer #2:

Remarks to the Author:

Overall, I found this to be an interesting and important article that was well done. Evaluating various policies designed to reduce technology use, particularly when they potentially cross over into censorship, is a worthwhile endeavor. I have just a few comments.

First, where mentioning, in the first paragraph, the "diagnoses" offered by the APA and WHO, the authors should also note the widespread criticisms of the same (Thorsten, 2017, for instance for the APA and Aarseth et al., 2017 for WHO).

Also, and I might have missed this, but haven't there been evaluations of S. Korea's "shutdown" laws that have come to similar conclusions as this article. That might help to put these results into context as they are not so unexpected.

The methods look good and competently done to me.

My main critique is that the observed ORs are very small. Even where they are "statistically significant", an OR of 1.14, for instance, is simply too small to conclude there is any meaningful difference. Such an effect is as likely to be due to methodological noise as anything else (see Ferguson & Heene, 2021, and a OR of 1.14 is clearly "noise" level). The authors preregistered an OR of 2.0 and were correct to do so. They should stick to it. Interpreting the results as there having been no meaningful change in gamer behavior before or after the Chinese law is just as interesting (and leads us to the same place) as saying the Chinese law increased gaming. Scholars making a big fuss over trivial effects (and advancing self-serving arguments why it was ok to do so) was bad science when moral panic scholars did it...let's learn from that and not simply repeat the same mistake. You've got a null result...and that's ok! It's actually still super important in this context!

Signed,
Chris Ferguson

Reviewer #3:
Remarks to the Author:
Key results

This study examines the effects of China's September 2019 gaming regulation that restricted gaming hours for youth (under age 18) to 1.5 hours each day, 3.5 hours on public holidays, and set restricted play from 22:00 to 08:00. The authors exploit a unique dataset from Unity containing telemetry data over a 22 week timespan (16 August 2019 to 16 January 2020). They analyzed the data in two different ways using GAMs. First, they measured the simple prevalence of heavy play before and after the regulations. Second, they estimated whether individuals played more/less heavily after the restriction was enacted. The authors find that evidence that this policy implementation is associated with increased heavy gaming among adults. I particularly appreciated the discussion on mechanisms and thought the explanations of player and industry noncompliance to be particularly compelling.

Validity

As the authors note, the generalizability of the study is questionable due to the lack of understanding the age groups that the data represent as well as the fact that the data are limited to only Unity games which may or may not represent the market. I find the lack of ages (due to the agreement with Unity) frustrating as it prevents the first order effects from being studied (e.g. the effect of the curfew

on youth gaming utilization).

It would be good to rule out holidays as a potential source of the increase in heavy utilization.

Why would effects in adults necessarily show that the policy is ineffective? If it reduces the youth utilization substantially more than it increases adult utilization, then it could be argued that it is effective. Essentially, without an understanding of which users are in which age groups, I don't think we can state whether the policy was effective since it was a policy aimed at youth not adults. I would caution the authors to take care in not overstating the policy implications of these analyses, but to maintain a position that there are probably spillover effects into the adult market that are consistent with the policy being ineffective.

Originality and significance

The authors claim that the policy implementation resulted in an increase in adult heavy-gaming utilization and that it had no "wide-spread and uniform effect." I would suggest that this should read, "no wide-spread and uniform decrease in utilization."

The authors claim that this policy resulted in an increase in heavy-gaming among adults.

The results are consistent with the policy being ineffective and advance our understanding of these types of policies in the gaming space. However, I think the lack of causal inference and the limited descriptives regarding the study population might limit the general appeal.

Data & methodology

Data is restricted due to the data agreement with the company providing the data so I will not comment on its validity. Notably, the scale and novelty of the dataset lends to the overall credence of the paper as it is not analysed by the game company itself.

Preregistration

The authors deviated from their preregistration. I found the explanation to be satisfactory.

Appropriate use of statistics and treatment of uncertainties

These are appropriate.

Custom code

The code is available, but without the data it is difficult to assess the code.

Conclusions

The overall conclusion that the policy is not having much effect is fair given that the bulk of gaming is happening on adult accounts assuming that this study's demographics is similar to other similar studies' demographics.

If holidays are ruled out as a possible mechanism, I think the conclusion that the policy implementation is correlated with an increase in adult utilization is fine.

Whether or not the policy is effective, cannot be answered with this data. That being said, as a first foray into the effects of this policy, the results are consistent with the policy being ineffective.

Suggested improvements

Rule out holidays/time off as a source of the increased utilization.

Given that this paper is essentially descriptive in nature, I would have really liked to see a lot more descriptive statistics regarding the study population. If there was any way to explore youth vs. adult heterogeneity, that would go a long way. Perhaps, the authors could stratify based on types of games that might generally go along the youth/adult age ranges. Even just getting overall demographic statistics (e.g. age groups, gender, etc.) from Unity on their userbase would help frame the discussion.

I'm not sure it's possible, but if there was a way to track account creations, that might be of use. If there was a spike in account creations immediately post policy, one could make the argument that youth were creating accounts.

References: Does this manuscript reference previous literature appropriately? If not, what references should be included or excluded?

Yes.

Clarity and context

For the introduction, I would suggest that a paragraph be inserted after the first paragraph that discusses the question(s) the authors are addressing and the specific policy the paper examines and maybe a sentence or two on the data. It's nice to have the goals of the paper up front so that as we're reading, we know where we're going. It felt a bit like I was searching for the point of the paper. The abstract is great, but the intro should be able to stand on its own without the abstract

It's not clear from the text why the authors believe that the youth effect is underestimated. If the youth population is about 6% of the sample, this means the first stage effects of the policy change should affect the tails of the distribution. So any estimates of effects on the full population which is primarily adults do not necessarily reflect what is happening with the youth. Perhaps clarifying what is meant with "underestimated" is sufficient, or perhaps the authors meant that the increases in adult utilization is underestimated. As it stands, I think "underestimated" is incorrect because the youth

effects could be negative, positive, or neutral and we have no idea what the magnitude is.

I would also caution the authors to avoid implying causality and to rely heavily on the fact that the evidence suggests limited effectiveness of the policy change. The abstract strikes the right balance, but occasionally the authors overstate the implications of the evidence in my opinion. For example in lines 252-253, I would suggest altering the language slightly: "This paper suggests there might be a more fundamental underlying issue with such policies..."

The authors write that the policy implementation resulted in an increase in adult heavy-gaming utilization and that it had no "wide-spread and uniform effect." I would suggest that this should read, "no wide-spread and uniform decrease in utilization."

Please indicate any particular part of the manuscript, data, or analyses that you feel is outside the scope of your expertise, or that you were unable to assess fully.

The data is unavailable to assess due to the data agreement; thus, the data and the corresponding code are difficult to assess.

Author Rebuttal to Initial comments

Reviewer #1 Comments

You said: "This manuscript looks at Chinese video game play before and after a policy restricting youth use, and finds a paradoxical increase in total play after the policy came into place. The manuscript is well-motivated via discussion of issues in (youth) consumer protection in similar risky domains, and is an inventive use of mass real world data. My comments are as follows

p2 lines 70-85 There is discussion here of in my view two separate cases. You can have policies that simply do not work effectively and which allow risky behaviour to continue (e.g., US cigarette limitations). Then you have a policy that backfires by driving users toward more risky content (e.g., restrictions on conventional online gambling leading to users going to crypto gambling sites). I believe this second case is more relevant to this work here. The pornography example given on line 224 in the Discussion is a nice example of a policy backfiring, as the present results appear to show, and I think that would be a better motivator than e.g. the loot boxes example given here (which is an example of a policy merely being ineffective)."

Our response: First of all, we wish to thank the reviewer for their thorough and thoughtful comments. This is a really good conceptual point, and something that was not at all clear in our original manuscript. We have adjusted the framing of the manuscript to clearly differentiate these different cases.

More specifically, we have written: “These questions reflect a broader set of concerns regarding the regulation of online behaviour and consumption amongst young people in general. To begin with, there are concerns that some specific policies may not work effectively and hence allow potentially harmful activities to continue unabated. For example, e-cigarette sales are prohibited to minors in all states within the USA, yet such products are still known to be widely purchased online by youth⁴⁷. Additionally, there are concerns that other policies may ‘backfire’, and lead to the accidental emergence of novel sources of harm. For example, narratives around ‘black market’ gambling deal with the idea that overly-stringent regulation of gambling may drive individuals towards unregulated, and potentially riskier, spaces such as cryptocurrency-based gambling sites^{48–50..}”.

We have also substantially expanded the discussion to deal with this specific tension between a lack of effectiveness and an overt ‘backfire’. We believe that the analyses requested by yourself and the other reviewers suggest that the most plausible explanation for our data is that Chinese policy was ineffective, rather than it actually backfired and caused increased heavy gaming.

Rather than deal with this here, we instead talk about this throughout the rest of our response, under the specific points where analyses are called for by reviewers. In particular, we feel that the difference-in-difference analyses you request provide good evidence that claiming a backfire here would be incautious. We deal with this point immediately below.

You said: “There is some discussion of other potential drivers of the effect (e.g., national holidays), but I think this type of econometric-type analysis should go further to increase confidence in the results. For example, the Authors could compare against a culturally-similar country which did not see the policy change. Could a difference-in-difference analysis be conducted with for example Taiwan as the control case in order to better convince that the policy change is driving this overall trend?”

Our response: Again, this is a great idea for further analysis and we were excited to implement this suggestion. We have implemented difference-in-difference analyses comparing China with a variety of other territories: not just Taiwan, but the 50 territories in Unity's data with the most players (including Taiwan). This has led to substantial expansion of the manuscript: such analyses strongly support the case that we should not interpret our results as a 'backfire' (increase in heavy gaming), but rather as a potentially ineffective policy which fails to reduce heavy gaming. We reproduce below relevant additional material within the results and discussion which goes with these analyses.

In the results: "we conducted 'difference-in-difference' analyses with a variety of global territories to examine whether the observed increases in heavy gameplay in our dataset were unique to China or exceptional. As noted above, play in China tended to be more likely to be heavy after regulation (OR=1.14). However, during the same time period, similar or greater differences were observed in a variety of territories, ranging from Russia to Australia (see Figure 3 below)."

In the discussion:

"Lack of evidence for backfiring effect

One potential interpretation of the observed effect is that it represents a policy backfire: a situation in which Chinese policy was not simply ineffective, but in fact actively exacerbated the phenomenon it was attempting to suppress. We do not believe the data obtained here are consistent with such an explanation.

Firstly, the effect observed here failed to exceed our preregistered threshold for a practically meaningful effect and thus is inconsistent with a policy backfire. It is important to note that a sceptic may suggest that our preregistered odds ratio (2.0) may be overly conservative; that rigid cut-offs should not be followed unthinkingly; and that factors such as quality of measurement may influence the true size of a practically significant effect⁶⁴. However, importantly, our sensitivity analyses reveal that the heaviness of playtime within any country may potentially be typified by a degree of volatility that is consistent with the odds ratio observed for China in this case, rendering these arguments powerless. If one assumes that a country tends to play with completely stable levels of heaviness from one period to another, our observed odds ratio between pre-regulation China and post-regulation China may appear to reveal an important increase in heaviness of play. However, if countries routinely play more or less heavily to

approximately this extent even in the absence of any formal regulation then this statistic provides weak support to such interpretations. Sensitivity analyses suggest that increases in heavy playtime of the sort observed in China during this period are routinely seen across the globe (see Figure 3). For example, countries as diverse as Belgium, the Ukraine, Poland, Russia, and Sweden all observed greater increases in heavy play during the post-regulatory period than China did (all odds ratios >1.14). When viewed in this light, it is difficult to interpret the odds ratio observed here as of practically meaningful magnitude.”

You said: “p2 line 91 The thing is, if you dig into the UK child gambling stats then most children gambling are either doing so legally or with parental approval. For example, in this wave the national lottery could be used legally by 16 year olds (the law changed in 2020), and also private gambling between friends is legal for children to do. Instances when a child does gamble online are rare, and are overwhelmingly cases where they are doing so with explicit parental approval. This does make me think that parents giving their IDs to their children in order to continue gaming may well be a driver of these effects. The authors mention this a little but do not provide a compelling case about why this is not driving the effect. Overall, I think part of the observed effect might be due to the binarization of the outcome (gamers categorised either as heavy gamers or not). Some children might play exactly the same amount before/after the policy, but if they tend to start spreading their play across multiple IDs AFTER the policy, then this may make it look like there is a greater number of heavy gamers in the population as defined in this binary sense by the authors. Some analysis treating heavy play as a continuous variable would I think be needed to counter this argument.”

Our response: Again, this is a helpful point and something which we were keen to bring into the paper itself. We have run two separate analyses that tackle these issues. Firstly, we have considered playtime as a continuous variable and investigated mean playtime per account rather than whether an account is playing ‘heavily’ or not. This has led to significant additions to the manuscript.

In the results: “Next, in order to more closely test any possible confounding effect of binarising our outcome on our results, we treated playtime as a continuous variable. We examined whether the mean weekly playtime for a randomly selected account in a post-regulation week still tended to be higher than a randomly selected account in a pre-regulation week. This was the case: after playtime, accounts tended to play for marginally more hours each week (pooled mean=1.64 vs 1.76), suggesting that the outcomes reported above are not well-explained as a confounded product of our binary measurement scheme alone (see Figure 4).”

In the discussion: “A second possible explanation for the lack of reduction in heavy play observed here relates to a situation in which real reductions in playtime are confounded due to the binary classification scheme that we employ here. However, results of a sensitivity analysis comparing mean playtime for each week in our data was unable to provide evidence for such confounding: even when measured as a continuous variable, no reduction in playtime was observed following regulation (See Figure 4).”

Secondly (in response to a point by reviewer 3), we have analysed whether regulation led to a spike in new account identifiers being created for different games, which would be consonant with the scenario you outline above as well. We reproduce the relevant text for this later on in our response, under Reviewer 3’s specific comment. In short, we see no evidence for a spike in account creations after regulation came in.

We wish to again thank Reviewer 1 for their considerate and rigorous review. It was interesting to implement the additional analysis, and we hope that they are in agreement with us that they massively improve the quality of the paper. In particular, the difference-in-difference analysis was a lovely and very illuminating approach.

Reviewer #2 Comments

You said: “Overall, I found this to be an interesting and important article that was well done. Evaluating various policies designed to reduce technology use, particularly when they potentially cross over into censorship, is a worthwhile endeavor. I have just a few comments.

First, where mentioning, in the first paragraph, the "diagnoses" offered by the APA and WHO, the authors should also note the widespread criticisms of the same (Thorsten, 2017, for instance for the APA and Aarseth et al., 2017 for WHO).”

Our response: Many apologies - we did not intend to present ‘only one side’ of this story, and are aware of the criticisms of such diagnoses. We have augmented the manuscript’s introduction to make the lack of consensus here as clear as possible.

More specifically, our manuscript now contains the following text: “The validity of conditions such as gaming disorder (ICD-11) and Internet Gaming Disorder (DSM-V) is heavily contested. The potential codification of Internet Gaming Disorder in the DSM attracted substantial controversy and criticism^{10,11}. Similarly, the World Health Organisation’s decision to add gaming disorder into the ICD-11 has led to widespread debate amongst academics regarding its appropriateness^{12,13}. Indeed, research on dysregulated gaming in general is characterised by a lack of consensus.”. We have also inserted the suggested references into the manuscript within this subsection.

You said: “Also, and I might have missed this, but haven't there been evaluations of S. Korea's "shutdown" laws that have come to similar conclusions as this article. That might help to put these results into context as they are not so unexpected.”

Our response: Again, this is a good point and one that we are keen to address. We briefly touched on South Korea in our introduction but did not deal with it for more than a couple of words. However, we would argue that literature on the impact of South Korea’s shutdown laws, whilst useful, suffers from a reliance on self-report of time rather than the use of raw behavioural data. We initially noted this only in our introduction (“ A self-report data analysis suggested that South Korean playtime restrictions did not reduce playtime. However, crucially, such data are not based on raw behavioural estimates of playtime, and therefore may be prone to error.”)

We have augmented our discussion to deal with this. More specifically, we have added the following content: “This paper also forms a methodological blueprint for empirical investigation of how a broad range of regulatory measures may affect the technology sector: for example, the impact of the repeal of South Korea’s restrictive Cinderella law, and the effects of attempted regulation of loot boxes in Belgium.”

You said: “The methods look good and competently done to me.

My main critique is that the observed ORs are very small. Even where they are "statistically significant", an OR of 1.14, for instance, is simply too small to conclude there is any meaningful difference. Such an effect is as likely to be due to methodological noise as anything else (see Ferguson & Heene, 2021, and a OR of 1.14 is clearly "noise" level). The authors preregistered an OR of 2.0 and were correct to do so. They should stick to it. Interpreting the results as there

having been no meaningful change in gamer behavior before or after the Chinese law is just as interesting (and leads us to the same place) as saying the Chinese law increased gaming. Scholars making a big fuss over trivial effects (and advancing self-serving arguments why it was ok to do so) was bad science when moral panic scholars did it...let's learn from that and not simply repeat the same mistake. You've got a null result...and that's ok! It's actually still super important in this context!

Signed,

Chris Ferguson”

Our response: We find this point of view convincing. We are even more convinced after running the “difference-in-difference” analyses suggested by R1: They lend a lot of credibility to your statement here that what we are seeing is “clearly noise level”. In fact, any previous overinterpretation of odds ratios presents quite a nice case study for why preregistration works and sensible cut-offs for effect sizes are also a good idea. Our previous hesitation here was due to an uncomfortableness with the use of specific thresholds/cutoffs for effect sizes in general, and a feeling that these should be interpreted in a more contextual manner. We will stick to our preregistered OR and not overinterpret the null effect, whilst attempting to bring out this sentiment in our discussion. We have revised the paper throughout to make this as explicit as possible. This is in line with both editorial comments, and R3’s suggestion that our results are best interpreted as our data being consistent with an ineffective policy. For example, we have added subsections into the discussion that make this as clear as possible (e.g. “No reduction in heavy gaming observed following playtime mandates” / “Lack of evidence for backfiring effect”). We have also added substantial additional text to the manuscript that deals with this idea. We have quoted some of it above in our response to R1 – whilst it risks belabouring the point, we have selected a subset to reproduce below that we hope shows that we agree and endorse the views you outline above: “One potential interpretation of the observed effect is that it represents a policy backfire: a situation in which Chinese policy was not simply ineffective, but in fact actively exacerbated the phenomenon it was attempting to suppress. We do not believe the data obtained here are consistent with such an explanation.

Firstly, the effect observed here failed to exceed our preregistered threshold for a practically meaningful effect and thus is inconsistent with a policy backfire. It is important to note that a sceptic may suggest that our preregistered odds ratio (2.0) may be overly conservative; that rigid cut-offs should not be followed unthinkingly; and that factors such as quality of measurement may influence the true size of a practically significant effect⁶⁴. However, importantly, our sensitivity analyses reveal that the heaviness of playtime within any country may potentially be typified by a degree of volatility that is consistent with

the odds ratio observed for China in this case, rendering these arguments powerless. If one assumes that a country tends to play with completely stable levels of heaviness from one period to another, our observed odds ratio between pre-regulation China and post-regulation China may appear to reveal an important increase in heaviness of play. However, if countries routinely play more or less heavily to approximately this extent even in the absence of any formal regulation then this statistic provides weak support to such interpretations. Sensitivity analyses suggest that increases in heavy playtime of the sort observed in China during this period are routinely seen across the globe (see Figure 3). For example, countries as diverse as Belgium, the Ukraine, Poland, Russia, and Sweden all observed greater increases in heavy play during the post-regulatory period than China did (all odds ratios >1.14). When viewed in this light, it is difficult to interpret the odds ratio observed here as of practically meaningful magnitude.”

Again, thank you for the thoughtful and thorough review.

Reviewer #3 Comments

You said: This study examines the effects of China’s September 2019 gaming regulation that restricted gaming hours for youth (under age 18) to 1.5 hours each day, 3.5 hours on public holidays, and set restricted play from 22:00 to 08:00. The authors exploit a unique dataset from Unity containing telemetry data over a 22 week timespan (16 August 2019 to 16 January 2020). They analyzed the data in two different ways using GAMs. First, they measured the simple prevalence of heavy play before and after the regulations. Second, they estimated whether individuals played more/less heavily after the restriction was enacted. The authors find that evidence that this policy implementation is associated with increased heavy gaming among adults. I particularly appreciated the discussion on mechanisms and thought the explanations of player and industry noncompliance to be particularly compelling.

Validity

As the authors note, the generalizability of the study is questionable due to the lack of understanding the age groups that the data represent as well as the fact that the data are limited to only Unity games which may or may not represent the market. I find the lack of ages

(due to the agreement with Unity) frustrating as it prevents the first order effects from being studied (e.g. the effect of the curfew on youth gaming utilization).

It would be good to rule out holidays as a potential source of the increase in heavy utilization.

In order to rule out public holidays as a potential source for the effect (or lack thereof!) observed here, a team member who is fluent in Chinese has formally reviewed relevant official documentation from the period.

Additions to our methods as a result of this are as follows: “As noted in our literature review, China’s 2019 playtime mandate is characterised by permitting different hours of play on days that are public holidays and those are not. On public holidays, minors may play for 3hrs per day – double the 1.5hrs that is permissible on other days. This presents a potential confound: If public holidays occur more frequently in the period after regulation, they may artificially inflate playtime for this period, leading researchers to either inappropriately conclude that the policy was ineffective, or that it may have even increased the prevalence of heavy play. In order to investigate the potentially confounding effects of public holidays, a member of the research team who is fluent in Chinese I examined official government sources regarding public holidays during this period^{70,71}. During the period under analysis, the following dates were designated as public holidays: 13th September 2019 (Dragon Boat Festival); 1-7 October 2019 (National Golden Week); 1st January 2020 (New Year’s Day). Of these, only 1st January 2020 falls after regulation. In addition to this, two specific weekend dates prior to regulation (September 29th, October 12th) were used as workdays to allow for the National Golden Week of uninterrupted holidays. Thus, there were 6 public holiday days prior to regulation; and one following regulation.”

The relevant subsection in our discussion now reads: “A first possible explanation is confounding due to history effects: if more public holidays occurred in the period under study after regulation than the period before regulation, heavy playtime due to public holidays may confound results. However, 5 more public holidays (6 days: one for the Dragon Boat Festival and five days for the National Golden Week) fell in the period before the restrictions than after (1 day: just New Year's Day), rendering this possible explanation inert. After a systematic search of relevant Chinese-language literature regarding holidays (see ‘Methods’) we are aware of no other obvious history effects that may potentially confounding results in such a manner.”

You said: “Why would effects in adults necessarily show that the policy is ineffective? If it reduces the youth utilization substantially more than it increases adult utilization, then it could be argued that it is effective. Essentially, without an understanding of which users are in which age groups, I don’t think we can state whether the policy was effective since it was a policy aimed at youth not adults. I would caution the authors to take care in not overstating the policy implications of these analyses, but to maintain a position that there are probably spillover effects into the adult market that are consistent with the policy being ineffective.”

Our response: Thank you for this point. Following both your remarks (and aligned feedback from R1 and R2) we are in complete agreement that the most correct interpretation of our results are that the data appear consistent with the policy being ineffective. We have revised our language throughout to follow the exemplar you give above, and have removed any and all direct claims about policy implications (e.g. that the policy caused heavier play).

You said: “Originality and significance

The authors claim that the policy implementation resulted in an increase in adult heavy-gaming utilization and that it had no “wide-spread and uniform effect.” I would suggest that this should read, “no wide-spread and uniform decrease in utilization.”

The authors claim that this policy resulted in an increase in heavy-gaming among adults.

The results are consistent with the policy being ineffective and advance our understanding of these types of policies in the gaming space. However, I think the lack of causal inference and the limited descriptives regarding the study population might limit the general appeal.”

We are in complete agreement with these points. In particular, we agree with the statement regarding the limited causal inferences we can make here, and agree with the reviewer that our data are better interpreted as consistent with an ineffective policy. We have revised this detail throughout, most notably in the title of the paper, which now reads: “Chinese playtime mandates did not reduce heavy gaming in one segment of the video games industry”

You said: "Data & methodology"

Data is restricted due to the data agreement with the company providing the data so I will not comment on its validity. Notably, the scale and novelty of the dataset lends to the overall credence of the paper as it is not analysed by the game company itself.

Preregistration

The authors deviated from their preregistration. I found the explanation to be satisfactory.

Appropriate use of statistics and treatment of uncertainties

These are appropriate.

Custom code

The code is available, but without the data it is difficult to assess the code.

Conclusions

The overall conclusion that the policy is not having much effect is fair given that the bulk of gaming is happening on adult accounts assuming that this study's demographics is similar to other similar studies' demographics.

If holidays are ruled out as a possible mechanism, I think the conclusion that the policy implementation is correlated with an increase in adult utilization is fine.

Whether or not the policy is effective, cannot be answered with this data. That being said, as a first foray into the effects of this policy, the results are consistent with the policy being ineffective.

Suggested improvements

Rule out holidays/time off as a source of the increased utilization.

Our response: As described in response to an earlier comment, we have augmented our methods with information which we believe effectively rules out public holidays as an explanation for the increased utilization (or rather - lack of decrease in utilization). In addition to this, we have (described above) revised our language throughout the paper to refer to the results as being consistent with an ineffective policy, and have tried wherever possible to not use causal language.

You said: “Given that this paper is essentially descriptive in nature, I would have really liked to see a lot more descriptive statistics regarding the study population. If there was any way to explore youth vs. adult heterogeneity, that would go a long way. Perhaps, the authors could stratify based on types of games that might generally go along the youth/adult age ranges. Even just getting overall demographic statistics (e.g. age groups, gender, etc.) from Unity on their userbase would help frame the discussion.”

Our response: This is one of the main limitations of the work conducted here, and the problematic nature of it is something we have attempted to bring out in our conclusions in response to the reviewer’s concerns. In essence, the problem is that this vast secondary dataset has been gathered by Unity for totally different purposes than the scientific ones we are utilising it for here, and thus it lacks the ideal information to inform scientific research. If we were collecting this gigantic dataset from scratch, we would incorporate rich demographic collection through processes of data fusion. If this limitation is not remediated, significant future work will also cause similar frustrations. Future work must focus on the generation of data infrastructure that allows the scientific community to access data like this - but with the kind of metadata we need for our work (e.g. age information). We have augmented the manuscript’s discussion with text dealing with this matter: “In our data, playtime appeared marginally (but not importantly) heavier post-regulation. However, it is crucial to note that we lack age information for each player in our dataset. This lack of metadata means that we cannot test whether unequal processes may

be in operation simultaneously within the population under observation: for example, we cannot falsify the idea that an increase in heavy gaming amongst adults could be co-occurring with youth simultaneously playing less heavily. This lack of relevant demographic detail is a key limitation of the use of large-scale industry datasets such as the one employed here⁶³. We maintain that the result observed here is most parsimoniously explained by an ineffective policy. Nonetheless, in order to build on this work, future research must focus on generating data infrastructure: technological frameworks that allow independent access to large-scale behavioural data fused to relevant self-report or demographic indicators”

You said: “I’m not sure it’s possible, but if there was a way to track account creations, that might be of use. If there was a spike in account creations immediately post policy, one could make the argument that youth were creating accounts.”

Our response: This is a great suggestion and we thank you for it. We have negotiated access to installation data from Unity for the period in question, and report in the paper a sensitivity analysis over this data: “Finally, we examined a suggestion by a reviewer that we could track account creations in order to indirectly assess whether youth were creating additional accounts. In order to do so, we examined whether there was a spike in account creations after regulation came into place. We were unable to observe such a change: if anything, fewer accounts were created on each of the weeks following regulation when compared to the 11 weeks preceding the playtime mandates coming into effect (see Figure 6, 127.33m vs 118.57m).”

You said: “References: Does this manuscript reference previous literature appropriately? If not, what references should be included or excluded?”

Yes.

Clarity and context

For the introduction, I would suggest that a paragraph be inserted after the first paragraph that discusses the question(s) the authors are addressing and the specific policy the paper examines and maybe a sentence or two on the data. It’s nice to have the goals of the paper up

front so that as we're reading, we know where we're going. It felt a bit like I was searching for the point of the paper. The abstract is great, but the intro should be able to stand on its own without the abstract"

Thank you for the kind words! We have done as suggested and inserted the following text to the manuscript within the first paragraph:

"In the wake of these concerns, a variety of governments have considered regulatory measures aimed at reducing playtime, particularly amongst young people. The most radical of these was enacted in mainland China in 2019: The Notice on the Prevention of Online Gaming Addiction in Juveniles mandated that individuals aged under 18 played no more than 1.5hrs each day (or 3hrs on public holidays) 6. Despite the importance of this regulation, its effectiveness has previously been impossible to establish due to a lack of large-scale behavioural data regarding playtime in China. Here we use approximately 7bn hours of playtime data drawn from mainland China in the weeks preceding and following the implementation of playtime mandates to investigate whether these regulations were effective in reducing heavy gaming."

You said: "It's not clear from the text why the authors believe that the youth effect is underestimated. If the youth population is about 6% of the sample, this means the first stage effects of the policy change should affect the tails of the distribution. So any estimates of effects on the full population which is primarily adults do not necessarily reflect what is happening with the youth. Perhaps clarifying what is meant with "underestimated" is sufficient, or perhaps the authors meant that the increases in adult utilization is underestimated. As it stands, I think "underestimated" is incorrect because the youth effects could be negative, positive, or neutral and we have no idea what the magnitude is."

Our response: Your point is extremely well-taken: We have removed the offending text from the manuscript (i.e. we have removed 'Thus, the figures generated here will naturally underestimate changes to the prevalence of heavy playtime amongst young people.'). This is in line with the more general revision of language throughout the text to tend to be more conservative.

You said: I would also caution the authors to avoid implying causality and to rely heavily on the fact that the evidence suggests limited effectiveness of the policy change. The abstract

strikes the right balance, but occasionally the authors overstate the implications of the evidence in my opinion. For example in lines 252-253, I would suggest altering the language slightly: “This paper suggests there might be a more fundamental underlying issue with such policies...”

Our response: We are in complete agreement with the reviewer, and apologise for any unintentional slips in our language. We have thoroughly revised the paper to remove any statements that may be interpreted as causal claims.

You said: “The authors write that the policy implementation resulted in an increase in adult heavy-gaming utilization and that it had no “wide-spread and uniform effect.” I would suggest that this should read, “no wide-spread and uniform decrease in utilization.”

Our response: We have adjusted the text in line with the reviewer’s recommendation. This is consistent with our general rewording of the entire paper to focus on the idea that our data are consistent with an ineffective policy.

You said: “Please indicate any particular part of the manuscript, data, or analyses that you feel is outside the scope of your expertise, or that you were unable to assess fully.

The data is unavailable to assess due to the data agreement; thus, the data and the corresponding code are difficult to assess.”

We thank the reviewers for the time and effort they put into assessing the manuscript. We hope they agree with us that the changes we have made in response to their reviews substantially improve both the paper itself, and the conclusions that it makes.

Decision Letter, first revision:

6th March 2023

Dear Dr Zendle,

Thank you once again for your revisions to the manuscript, entitled "Chinese playtime mandates did not reduce heavy gaming in one segment of the video games industry,"

Your manuscript has been evaluated by the same three reviewers, and you will see that they think the paper has improved in revision. Reviewer #3, however, suggests adding an additional analysis (see comments below), and I was wondering if you could do so before we accept the paper in principle?

[REDACTED]

Sincerely,

Arunas Radzvilavicius, PhD
Editor, Nature Human Behaviour
Nature Research

REVIEWER COMMENTS:

Reviewer #1:

Remarks to the Author:

The authors have responded very well to the feedback provided by the Reviewers. I believe this is an instance where publication should be supported based on the scale of the data, quality of analysis, and significance of the issue, over considerations regarding the "noteworthiness" of the findings.

Reviewer #2:

None

Reviewer #3:

Remarks to the Author:

I appreciate the revisions the authors implemented; especially the reframing of the results as showing a lack of policy effect. Considering the revisions, I have a few additional comments.

Major comments

Difference-in-differences is arguably a stronger methodology, and I think this section could be expanded to highlight the lack of findings. I think it would be good to see at least one event study as an example (probably China and Taiwan) as it would be a way to empirically address history effects as well as visually show the policy ineffectiveness. By showing that there is no difference in the pre- and post-trends between China and Taiwan, this should strengthen your argument around holidays and show that the policy effect is not being detected. A good example of this type of analysis and the appropriate figure can be found here: Miller, Sarah, Norman Johnson, and Laura R Wherry, "Medicaid

and mortality: new evidence from linked survey and administrative data,” Technical Report, National Bureau of Economic Research 2019.

Minor comments

I would also suggest that showing the event studies for each of your outcomes of interest would serve as a good robustness check and satisfy the causal inference folks. I think this would fit well in supplementary materials rather than the main text since it primarily appeals to causal inference researchers. I don’t expect it to change any results but will strengthen your paper.

Page 3, second paragraph under results: I would suggest deleting “notedly,” from “However, notably, ...”

There is a typo on page 6, paragraph 1; “centring” should be “centering”.

Page 9, paragraph starting with, “A first possible explanation...”: The final sentence should be split into two at the semicolon. “... rendering this possible explanation inert. After a systematic...”

Page 10, 2nd full paragraph: the semicolon should be removed from “This would account for both a lack of increase in account creations; and a lack of increase in total playtime in China”

Page 10, final sentence: the colon should be a period in “However, we would suggest caution in this interpretation: It is crucial to point out that the observed odds ratio in this study did not exceed our preregistered threshold for practical importance.”

Page 11, paragraph 3: the colon should be a period in “This paper also forms a methodological blueprint for investigating how a broad range of regulatory measures may affect the technology sector: for example, future analyses could focus on the impact of the repeal of South Korea’s restrictive Cinderella law, and the effects of attempted regulation of loot boxes in Belgium”

Author Rebuttal, first revision:

Reviewer #3 Comments

You said: “I appreciate the revisions the authors implemented; especially the reframing of the results as showing a lack of policy effect. Considering the revisions, I have a few additional comments.

Major comments

Difference-in-differences is arguably a stronger methodology, and I think this section could be expanded to highlight the lack of findings. I think it would be good to see at least one event study as an example (probably China and Taiwan) as it would be a way to empirically address history effects as well as visually show the policy ineffectiveness. By showing that there is no difference in the pre- and

post-trends between China and Taiwan, this should strengthen your argument around holidays and show that the policy effect is not being detected. A good example of this type of analysis and the appropriate figure can be found here: Miller, Sarah, Norman Johnson, and Laura R Wherry, "Medicaid and mortality: new evidence from linked survey and administrative data," Technical Report, National Bureau of Economic Research 2019."

Thank you very much for this useful comment. We have run event studies as recommended, comparing 1,000,000 randomly-selected observations within China against 1,000,000 randomly selected observations taken from within the East Asian cultural sphere (Taiwan, Japan, South Korea, Hong Kong, Vietnam). We have conducted these studies for outcomes in terms of both changes in mean playtime per week; and changes in the prevalence of heavy play. As you predicted, they show no plausibly important difference in pre-and-post trends. However, as you also point out, they do provide a good visual description of what this lack of impact looks like. Finally, we are keen to implement methods that individuals from disciplines different to our own will understand and trust. This has been a helpful addition to the paper.

Minor comments

You said: "I would also suggest that showing the event studies for each of your outcomes of interest would serve as a good robustness check and satisfy the causal inference folks. I think this would fit well in supplementary materials rather than the main text since it primarily appeals to causal inference researchers. I don't expect it to change any results but will strengthen your paper."

Our response: This is a good point – the event study write-up is quite bulky and putting it into the supplemental materials is an excellent idea (we are getting dangerously close to our word limit!). We have added some text into the manuscript pointing our audiences towards this: "A final suggestion for sensitivity analysis was to reanalyze our data using a formal event study paradigm, as this is commonly employed for causal inference in health economics research⁶². Event studies were conducted assessing the impact of regulations on individual accounts in China when compared to individual accounts from elsewhere in the East Asian cultural sphere. Difference in differences were assessed with outcomes represented both by increases in mean playtime per player and increases in the likelihood of heavy play. In all instances, analyses returned results that failed to show a statistically significant reduction in either mean playtime or the likelihood of heavy play after regulation. For the purposes of brevity, these analyses and their results are reported in our supplemental materials rather than the main body of the paper."

You said: "Page 3, second paragraph under results: I would suggest deleting "notedly," from "However, notably, ..."

There is a typo on page 6, paragraph 1; “centring” should be “centering”.

Page 9, paragraph starting with, “A first possible explanation...”: The final sentence should be split into two at the semicolon. “... rendering this possible explanation inert. After a systematic...”

Page 10, 2nd full paragraph: the semicolon should be removed from “This would account for both a lack of increase in account creations; and a lack of increase in total playtime in China”

Page 10, final sentence: the colon should be a period in “However, we would suggest caution in this interpretation: It is crucial to point out that the observed odds ratio in this study did not exceed our preregistered threshold for practical importance.”

Page 11, paragraph 3: the colon should be a period in “This paper also forms a methodological blueprint for investigating how a broad range of regulatory measures may affect the technology sector: for example, future analyses could focus on the impact of the repeal of South Korea’s restrictive Cinderella law, and the effects of attempted regulation of loot boxes in Belgium”

Thank you for spotting all of these – this is extremely helpful of you. We have made all of your suggested minor amendments.

Again, we would like to thank all reviewers and the editor for the extremely rigorous and pleasant review process.

Decision Letter, second revision:

13th April 2023

Dear Dr. Zendle,

Thank you for your patience as we’ve prepared the guidelines for final submission of your Nature Human Behaviour manuscript, “Chinese playtime mandates did not reduce heavy gaming in one segment of the video games industry” (NATHUMBEHAV-22113105B). Please carefully follow the step-by-step instructions provided in the attached file, and add a response in each row of the table to indicate the changes that you have made. Please also check and comment on any additional marked-up edits we have proposed within the text. Ensuring that each point is addressed will help to ensure that your revised manuscript can be swiftly handed over to our production team.

We would hope to receive your revised paper, with all of the requested files and forms within two-three weeks. Please get in contact with us if you anticipate delays.

Nature Human Behaviour offers a Transparent Peer Review option for new original research manuscripts submitted after December 1st, 2019. As part of this initiative, we encourage our authors to support increased transparency into the peer review process by agreeing to have the reviewer comments, author rebuttal letters, and editorial decision letters published as a Supplementary item. When you submit your final files please clearly state in your cover letter whether or not you would like to participate in this initiative. Please note that failure to state your preference will result in delays in accepting your manuscript for publication.

In recognition of the time and expertise our reviewers provide to Nature Human Behaviour's editorial process, we would like to formally acknowledge their contribution to the external peer review of your manuscript entitled "Chinese playtime mandates did not reduce heavy gaming in one segment of the video games industry". For those reviewers who give their assent, we will be publishing their names alongside the published article.

Cover suggestions

As you prepare your final files we encourage you to consider whether you have any images or illustrations that may be appropriate for use on the cover of Nature Human Behaviour.

ORCID

Non-corresponding authors do not have to link their ORCIDs but are encouraged to do so. Please note that it will not be possible to add/modify ORCIDs at proof. Thus, please let your co-authors know that if they wish to have their ORCID added to the paper they must follow the procedure described in the

following link prior to acceptance:

Nature Human Behaviour has now transitioned to a unified Rights Collection system which will allow our Author Services team to quickly and easily collect the rights and permissions required to publish your work. Approximately 10 days after your paper is formally accepted, you will receive an email in providing you with a link to complete the grant of rights. If your paper is eligible for Open Access, our Author Services team will also be in touch regarding any additional information that may be required to arrange payment for your article. Please note that you will not receive your proofs until the publishing agreement has been received through our system.

Please note that *Nature Human Behaviour* is a Transformative Journal (TJ). Authors may publish their research with us through the traditional subscription access route or make their paper immediately open access through payment of an article-processing charge (APC). Authors will not be required to make a final decision about access to their article until it has been accepted. Find out more about Transformative Journals

[REDACTED]

Best regards,
Alex McKay
Editorial Assistant
Nature Human Behaviour

On behalf of

Arunas Radzvilavicius, PhD
Senior Editor, Nature Human Behaviour
Nature Research

Reviewer #3:

Remarks to the Author:

I am grateful for the patient and thoughtful responses to my comments as well as those of the other reviewers. I also appreciate the considerable effort you have put into making the paper rigorous and well-supported, and I believe all the comments have been well addressed.

Thank you for taking the time to consider all the comments and for your work on this important topic.

Final Decision Letter:

Dear Dr Zendle,

We are pleased to inform you that your Article "No evidence that Chinese playtime mandates reduced heavy gaming in one segment of the video games industry", has now been accepted for publication in *Nature Human Behaviour*.

Please note that *Nature Human Behaviour* is a Transformative Journal (TJ). Authors whose manuscript was submitted on or after January 1st, 2021, may publish their research with us through the traditional subscription access route or make their paper immediately open access through payment of an article-processing charge (APC). Authors will not be required to make a final decision about access to their article until it has been accepted. IMPORTANT NOTE: Articles submitted before January 1st, 2021, are not eligible for Open Access publication. Find out more about Transformative Journals

Acceptance of your manuscript is conditional on all authors' agreement with our publication policies (see <http://www.nature.com/nathumbehav/info/gta>). In particular your manuscript must not be published

elsewhere and there must be no announcement of the work to any media outlet until the publication date (the day on which it is uploaded onto our web site).

With best regards,

Arunas Radzvilavicius, PhD
Senior Editor, Nature Human Behaviour
Nature Research

P.S. Click on the following link if you would like to recommend Nature Human Behaviour to your librarian
<http://www.nature.com/subscriptions/recommend.html#forms>

** Visit the Springer Nature Editorial and Publishing website at
www.springernature.com/editorial-and-publishing-jobs for more information about our career
opportunities. If you have any questions please click here.**

This email has been sent through the Springer Nature Tracking System NY-610A-NPG&MTS

Confidentiality Statement:

This e-mail is confidential and subject to copyright. Any unauthorised use or disclosure of its contents is prohibited. If you have received this email in error please notify our Manuscript Tracking System Helpdesk team at <http://platformsupport.nature.com>.

Details of the confidentiality and pre-publicity policy may be found here
<http://www.nature.com/authors/policies/confidentiality.html>

Privacy Policy | Update Profile